# Multiple short windows of calcium-dependent protein kinase 4 activity coordinate distinct cell cycle events during *Plasmodium* gametogenesis

Hanwei Fang[1†], Natacha Klages[1†], Bastien Baechler[1], Evelyn Hillner[2], Lu Yu[2], Mercedes Pardo[2], Jyoti Choudhary[2], Mathieu Brochet[1*]

[1]Department of Microbiology and Molecular Medicine, University of Geneva, Geneva, Switzerland; [2]Proteomic Mass-spectrometry Team, Wellcome Trust Sanger Institute, Cambridge, United Kingdom

**Abstract** Malaria transmission relies on the production of gametes following ingestion by a mosquito. Here, we show that $Ca^{2+}$-dependent protein kinase 4 controls three processes essential to progress from a single haploid microgametocyte to the release of eight flagellated microgametes in *Plasmodium berghei*. A myristoylated isoform is activated by $Ca^{2+}$ to initiate a first genome replication within twenty seconds of activation. This role is mediated by a protein of the SAPS-domain family involved in S-phase entry. At the same time, CDPK4 is required for the assembly of the subsequent mitotic spindle and to phosphorylate a microtubule-associated protein important for mitotic spindle formation. Finally, a non-myristoylated isoform is essential to complete cytokinesis by activating motility of the male flagellum. This role has been linked to phosphorylation of an uncharacterised flagellar protein. Altogether, this study reveals how a kinase integrates and transduces multiple signals to control key cell-cycle transitions during *Plasmodium* gametogenesis.

*For correspondence: Mathieu. Brochet@unige.ch

[†]These authors contributed equally to this work

Competing interests: The authors declare that no competing interests exist.

## Introduction

Malaria is caused by vector-borne protozoan parasites of the genus *Plasmodium* that cycle between mosquitoes and vertebrate hosts. Malaria pathology is linked to the proliferation of asexual blood stage parasites, whereas transmission to the mosquito is mediated by an obligatory sexual life cycle phase. Differentiation from asexually replicating stages into non-dividing male and female gametocytes takes place inside red blood cells. Following a period of maturation, the sexual precursors are available to initiate transmission when ingested by a mosquito. Gametocytes resume their development by responding to environmental signals including a small mosquito molecule, xanthurenic acid (XA), and a simultaneous drop in temperature (*Billker et al., 1998*).

Upon ingestion by a mosquito vector, *Plasmodium* male gametocyte undergoes explosive development. Within 10 min, it completes three rounds of genome replication followed by endomitosis within a single nucleus, assembles the component parts of eight axonemes, and escapes the red blood cell in a process called exflagellation. Circulating microgametocytes are arrested at a $G_0$-like stage of the cell cycle at the haploid level. After 15 s of induction by XA, eight basal bodies are assembled from a single microtubule organising centre (*Sinden et al., 1976*). After 1 min, the first genome replication is completed and the spindle of mitosis I is formed (*Billker et al., 2002*). At the same time each basal body nucleates one of the eight axonemes from large quantities of tubulin contained in the cytoplasm. By six minutes, the four spindles of mitosis III have formed and

**eLife digest** Malaria is caused by parasites called *Plasmodium*, which are carried between humans by infected mosquitoes. The parasite needs to move between its insect and human hosts to complete its life cycle, and efforts to stop the spread of malaria are now focussed on blocking this movement.

In the blood of infected humans, the parasite exists in a form known as a gametocyte, which can be male or female. When a mosquito bites and ingests the blood of a person with malaria, the gametocytes experience a sudden drop in temperature, which stimulates them to develop into sex cells. In less than ten minutes, each male gametocyte replicates its genetic material three times, divides this material into eight sex cells, and these new cells make lash-like appendages called flagella. The flagella allow the male sex cells to "swim" around the mosquito's blood meal in search of female sex cells.

Several signalling molecules help to control the development of male gametocytes. One such molecule is a protein known as CDPK4. Drugs targeting this protein inhibited the spread of a malaria parasite that infects rodents, yet a full understanding of how the CDPK4 protein controls the development of male gametocytes remains a mystery.

Here, Fang et al. reveal that in the ten minutes following a parasite entering a mosquito, CDPK4 controls three distinct events required for the development of male gametocytes. CDPK4 is required to stimulate the replication of the parasite DNA and to separate the duplicated genetic material into the new cells. CDPK4 influences both events within 30 seconds of the parasite sensing a drop in temperature, which was unexpected because these events are not usually directly connected in other organisms. Shortly after, CDPK4 is required to stimulate the sex cells' flagella to move.

Further experiments analysing the effect of CDPK4 on over 2,000 proteins show that CDPK4 controls the activity of multiple proteins in the parasite. Three of these proteins had not been characterized before and Fang et al. found that they are involved in critical stages leading up to the movement of the flagella.

Along with inhibiting the spread of *Plasmodium*, drugs that target CDPK4 can inhibit the spread of many related parasites, including the parasite that causes a disease called toxoplasmosis. These findings provide a first insight into how this protein works at a molecular level and may aid the development of more effective drugs in the future.

chromatin condensation only sets in at the end of mitosis III. In parallel gametocytes escape their host cell following the exocytosis of specialised secretory vesicles containing proteins with membranolytic activities. At the onset of exflagellation, axonemes become motile and swim out of the residual gametocyte body. As each basal body remains attached to a mitotic spindle pole, they drag a haploid genome that is incorporated into the exflagellating gamete.

Malaria parasites are highly divergent from model organisms and significant differences in the composition and properties of cell cycle regulators have been reported (*Gerald et al., 2011*). As a consequence little is known about how progression through the cell cycle is regulated in these parasites. Gametocyte stimulation by XA is followed by $Ca^{2+}$ mobilisation from internal stores after a lag phase of ~10 s (*Billker et al., 2004*) which requires active cGMP-dependent protein kinase G, PKG (*Brochet and Billker, 2016*; *Brochet et al., 2014*). In activated microgametocytes, the plant-like $Ca^{2+}$-dependent protein kinase 4 (CDPK4), which belongs to a family absent from the human genome, is required to enter S-phase in the rodent parasite *Plasmodium berghei* (*Billker et al., 2004*). Selective inhibitors of CDPK4 were shown to block exflagellation of *P. berghei in vivo* and of the human parasite *P. falciparum in vitro* placing CDPK4 as a promising drug target to reduce transmission of malaria (*Ojo et al., 2014*, *2012*).

Despite the importance of CDPK4 for transmission to the mosquito vector, its molecular functions remain unknown and none of its substrates have been identified. In this study, we took advantage of the highly synchronised nature of *P. berghei* gametogenesis to exactly identify when CDPK4 activity is required. By combining reverse and chemical genetics with molecular and cellular phenotyping,

we found that CDPK4 plays at least three distinct roles during male gametogenesis and we identified three effectors mediating each of these roles.

## Results

### A chemical genetic approach to modulate CDPK4 activity with a high time resolution

Small bumped-kinase inhibitors targeting *Plasmodium* CDPK4 were recently developed by capitalising on a small serine gatekeeper residue in the active site of the enzyme (*Ojo et al., 2014*, *2012*). One of these, compound 1294, was found to inhibit *P. falciparum* exflagellation through CDPK4 (*Ojo et al., 2014*). To ascertain for 1294 specificity in *P. berghei*, we set out to replace *cdpk4* with a drug-resistant allele, *cdpk4*$^{S147M}$-3xHA to prevent binding of 1294 to the ATP binding pocket (*Figure 1—figure supplement 1* parts 1 to 5). We however discovered that introduction of both a C-terminal 3xHA epitope tag and a S147M substitution imposed a fitness cost on CDPK4 function preventing exflagellation. Investigation of the effect of each single modification revealed exflagellation is delayed in CDPK4-3xHA but not in CDPK4$^{S147M}$ parasites. This indicates that a S147M substitution only imposes a minor cost that, on its own, does not impact on exflagellation. Compared to the parental wild-type strain, CDPK4$^{S147M}$ transgenic parasites showed a ~ 50 fold decrease in exflagellation susceptibility to 1294. Importantly, 1 µM 1294 blocked exflagellation by specifically targeting CDPK4 and this concentration was further used this study (*Figure 1A*).

### CDPK4 activity is required to initiate the first round of DNA replication, assemble the first mitotic spindle and initiate axoneme motility

To decipher how CDPK4 controls entry into S-phase during male gametogenesis, we first determined the kinetics of DNA replication (*Figure 1—figure supplement 1* parts 6 and 7). In non-activated parasites, around 85% of male gametocytes were haploid. A majority of diploid parasites could be observed between 30 s and 1 min after XA activation, while tetraploid microgametocytes were mainly detected between 2 and 5 min. The three rounds of genome replication seemed to be completed by 6 min when 75% of male parasites were octoploid (*Figure 1—figure supplement 1* part 8).

   We then investigated the requirement for CDPK4 activity to control genome replications by adding 1294 at different time points during gametogenesis (*Figure 1B*). Addition of 1294 prior to activation dramatically reduced the ability of male gametocytes to replicate their genome as previously described for CDPK4-KO parasites (*Billker et al., 2004*). We were however able to detect 25% of microgametocytes progressing to the diploid state but not further to the tetra- or octoploid levels. When 1294 was added 20 s after activation, no inhibition of DNA replication was observed. Altogether, this reveals that CDPK4 activity between 0 and 20 s post-activation is important for the 1N/2N and subsequent 2N/4N transitions but not for DNA replication per se.

   Interestingly, almost no diploid parasites reached the tetraploid state suggesting that CDPK4 plays a second role between the first and second rounds of genome replication. To test this hypothesis, parasites treated with 1294 or DMSO were imaged 1 min after activation, when the first mitotic spindle is observable (*Figure 1C*). In DMSO treated parasites, 64% of male gametocytes showed spindle-like structures while this proportion decreased to 7% in parasites treated with 1294 (*Figure 1D*) indicating that CDPK4 activity is also required for mitotic spindle formation.

   We further took advantage of the versatility of the chemical approach to test whether CDPK4 is required later during male gametogenesis. To do so, we added 1294 30 s after activation, a time point at which CDPK4 is not required to initiate DNA replication anymore. With such a treatment, parasites normally developed to the 8N state and assembled axonemes. However, DNA did not condense, axonemes remained around the nucleus and no exflagellation was observed (*Figure 1C and E*). A similar phenotype was observed when 1294 was added only seconds prior to the initial exflagellation events at 9 min post-activation (*Figure 1C and E–F*). However, when added at 15 min post-activation, 1294 did not inhibit flagellar beating (*Figure 1G*) or ookinete formation and motility (*Figure 1—figure supplement 1* parts 9 and 10). Altogether, this shows that CDPK4 activity seems to be redundant between 30 s and 9 min after activation but is further required at a late stage immediately preceding the onset of axoneme motility, DNA condensation and cytokinesis.

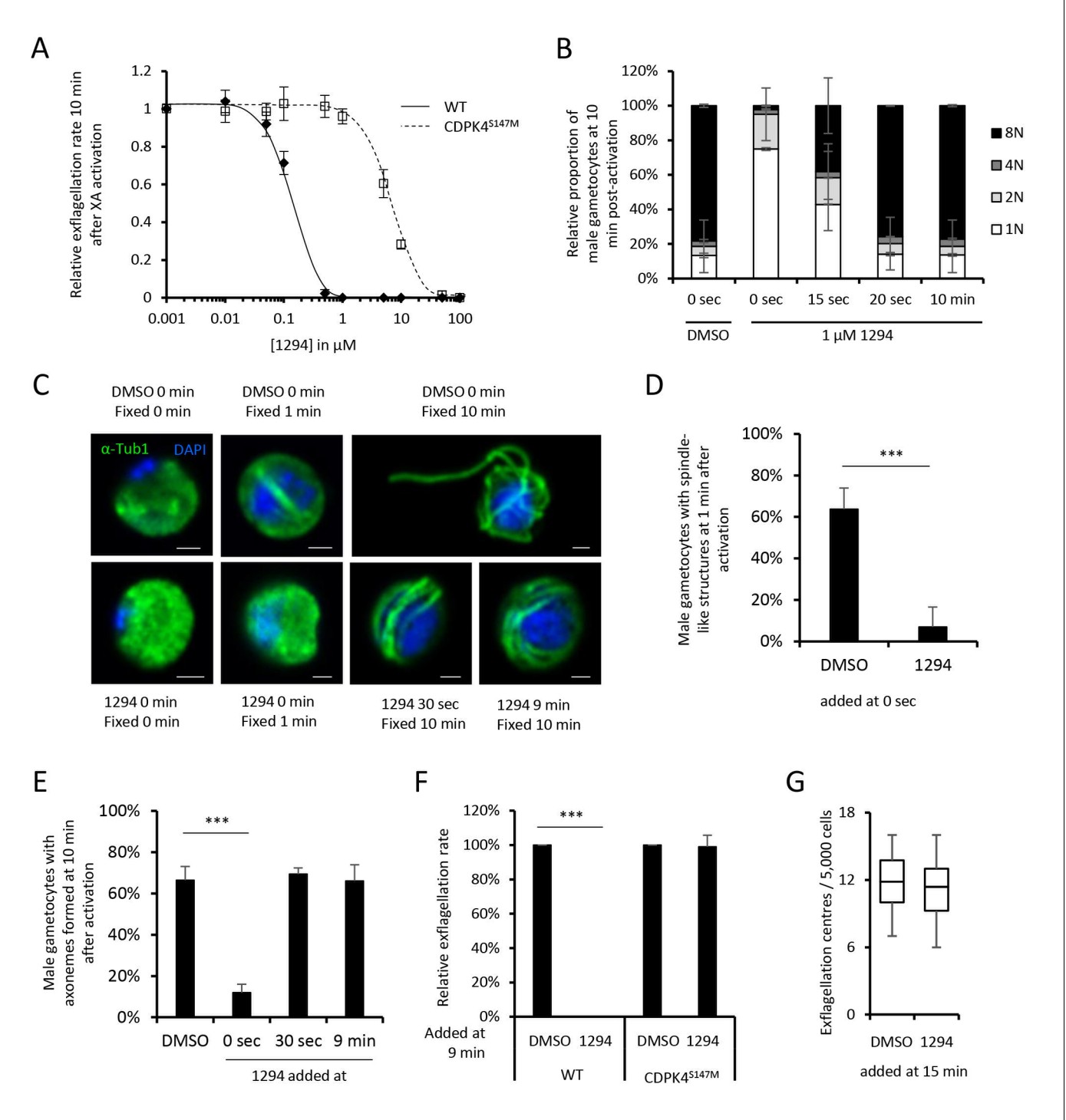

**Figure 1.** A chemical genetic approach reveals CDPK4 activity is required to initiate the first round of DNA replication, assemble the first mitotic spindle and initiate axoneme motility. (**A**) A line expressing a CDPK4$^{S147M}$ allele is 50 times more resistant to compound 1294. Error bars show standard deviations, n = 3. (**B**) Effect of 1 µM 1294 addition at multiple time points after XA activation on microgametocytes ploidy. CDPK4 activity is required between 10 and 20 s to initiate the first round of DNA replication and the second genome replication. CDPK4 activity is however not required for DNA replication per se. Error bars represent standard deviations, n = 2. (**C**) Immunofluorescence assays showing the effect of 1294 at different time points after activation. Addition of 1 µM 1294 at the time of activation prevents the formation of mitotic spindles as observed 1 min after activation in presence of DMSO. Addition of 1294 at 30 s or at 9 min post-activation does not inhibit axoneme formation but blocks initiation of axoneme motility and condensation of chromatin. Absence of 1294 treatment leads to the exflagellation of male gametes 10 min after activation. Scale bars = 1 µm. (**D**)

*Figure 1 continued on next page*

*Figure 1 continued*

Quantification of macrogametocytes showing mitotic spindles at 1 min post-activation in DMSO or 1294 treated parasites. Error bars show standard deviations, n = 2, *** Student's T-test, $p \leq 0.001$. (E) Quantification of macrogametocytes showing axonemes formed at 10 min after activation. Error bars show standard deviations, n = 2, *** Student's T-test, $p \leq 0.001$. (F) Quantification of exflagellation events in the WT and CDPK4$^{S147M}$ lines when DMSO or 1294 were added 9 min after activation. Error bars show standard deviations n = 2, *** Student's T-test, $p \leq 0.001$. (G) When added 15 min post-XA activation, 1294 does not inhibit motility of active microgametes, n = 3, *** Student's T-test, $p \leq 0.001$.

The following figure supplement is available for figure 1:

**Figure supplement 1.** CDPK4 is required to initiate the first round of DNA replication but not for ookinete development or motility.

## CDPK4 is part of the MCM complex in non-activated gametocytes and mainly interacts with cytoskeletal proteins in late gametogenesis

To gain further insights into how CDPK4 regulates these transitions, we first aimed at identifying its interacting proteins in non-activated gametocytes. A total of 150 proteins were immunoprecipitated in both CDPK4-3xHA and CDPK4-2xmyc lysates after cross-linking but not in a WT control (*Figure 2—figure supplement 1* parts 1 to 3 and *Supplementary file 1*). MCM2-7/Cdt1 and ORC proteins that are part of the pre-replicative complex represented the most enriched molecular components (*Figure 2A*). Proteins of the microtubule cytoskeleton and of the replisome were also enriched including polymerases α, δ, and ε, the proliferating nuclear antigen, replication factor C complexes and replication factor A.

As MCM2-7/Cdt1 represented a highly enriched and abundant protein complex immunoprecipitated with CDPK4 in non-activated gametocytes, we first confirmed this interaction by determining the partners of MCM5-3xHA in the same conditions (*Figure 2—figure supplement 1* parts 4 and 5). We found 103 proteins interacting with CDPK4-3xHA, CDPK4-2xmyc and MCM5-3xHA indicating that CDPK4 and MCM5 share an overlapping molecular environment (*Figure 2B*). Consistently, CDPK4 represented one of the most abundant interactors of MCM5-3xHA. Altogether this suggests that CDPK4 is part of the MCM2-7/Cdt1 complex and may regulate the pre-replicative complex to initiate DNA replication.

To look for putative molecular roles explaining the late requirement of CDPK4, we identified interacting partners of CDPK4-2xmyc and CDPK4-3xHA 10 min after activation (*Supplementary file 1*). Cytoskeletal proteins predicted to be components of the microgamete axonemes (*Talman et al., 2014*) were the most abundant and significantly enriched cellular components (p-value=$5.10^{-4}$). It is however important to note that most of this cellular component was also enriched in non-activated gametocytes and that cytoskeletal proteins are abundant in male gametocytes (*Khan et al., 2005*). It is currently unclear whether these proteins represent relevant or specific interactors. Nuclear proteins involved in DNA replication were nevertheless much less abundant compared to earlier time points (*Supplementary file 1*). Traces of CDPK4 were also found to be incorporated in exflagellated male gametes (*Figure 2—figure supplement 1* parts 4 and 5 - [*Talman et al., 2014*]) in accordance with a role in the initiation of axoneme activity.

## CDPK4 activity is required for the assembly of the pre-replicative complex

Assembly of the pre-replicative complex is a dynamic process (*Bell and Kaguni, 2013*). In eukaryotes, ORC1-6 proteins bind to origin sequences and associate with Cdc6. Cdc6 subsequently recruits the licensing factor Cdt1 and the heterohexameric MCM helicase complex. To investigate the dynamics of the CDPK4/MCM2-7/Cdt1 complex, we compared the CDPK4 interactome in non-activated gametocytes and in 15 second-activated gametocytes in the presence or absence of 1294. For 215 out of the 221 proteins immunoprecipitated with CDPK4-3xHA under all conditions, no obvious differences were detected between the three conditions (*Figure 1—figure supplement 1* part 8). However six proteins showed up to 8-fold increase in abundance upon gametocytes activation which was not observed when CDPK4 was inhibited (*Figure 2C*). A similar result was observed when using the CDPK4-2xmyc line (*Supplementary file 1* and *Figure 2—figure supplement 1* part 8). Among these proteins were ORC1, ORC2, and ORC5, while the three remaining proteins were not

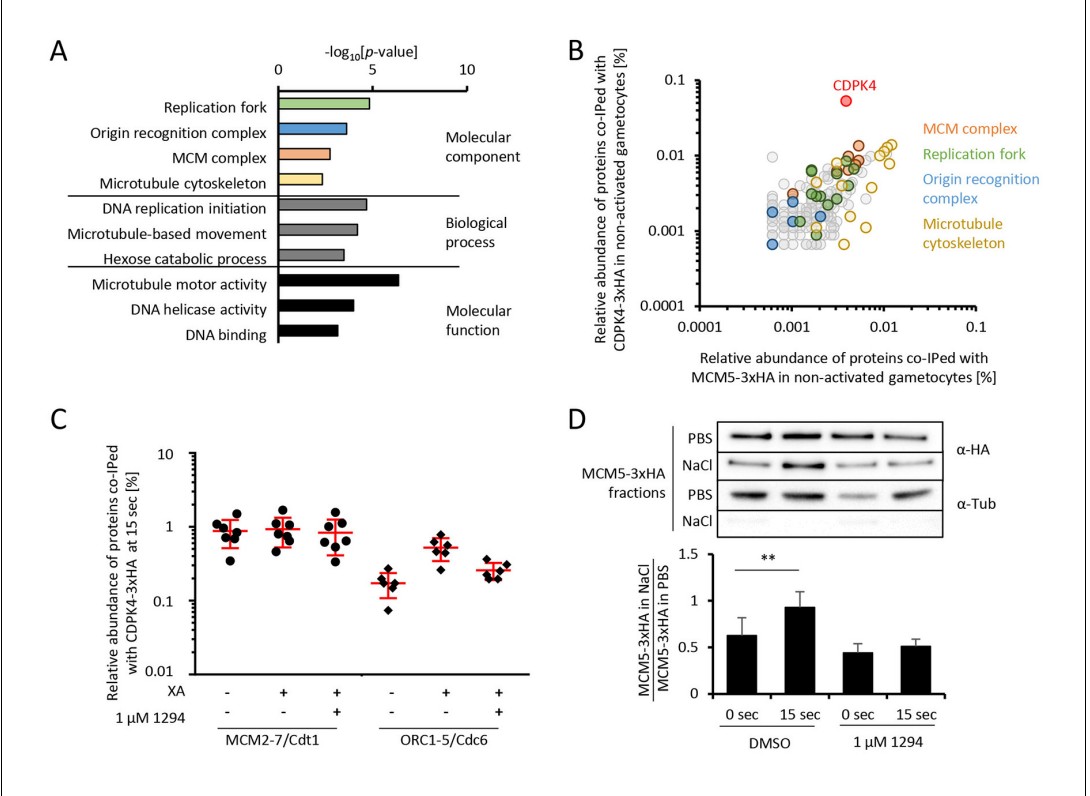

**Figure 2.** CDPK4 activity facilitates the assembly of the pre-replicative complex during microgametogenesis. (A) GO term enrichment analysis of proteins co-immunoprecipitated with CDPK4-3xHA in non-activated gametocytes reveals the kinase is interacting with proteins of the pre-replicative complex. Bonferroni corrected p-values are indicated. (B) 178 proteins are immunoprecipitated with both CDPK4-3xHA and MCM5-3xHA including all components of the ORC and the MCM complex. The relative abundance of proteins was determined as the number of spectral counts for each protein divided by the number of total spectral counts in the respective immunoprecipitate. (C) The interaction between CDPK4-3xHA and MCM2-7/Cdt1 proteins is stable during the first 15 s following XA stimulation while interaction with ORC1-5/Cdc6 proteins is increased when the kinase is activated. Data are representative of two independent biological replicates. (D) MCM5-3xHA is enriched in chromatin-enriched NaCl fractions at 15 s post-activation in gametocytes but not when 1294 is added; α-tubulin was used as a soluble control. Error bars show standard deviations, n = 3, ** Student's T-test, p≤0.01.

The following figure supplement is available for figure 2:

**Figure supplement 1.** CDPK4 activity is required to assemble the pre-replicative complex.

functionally annotated. Sequence analysis indicated that these three proteins likely correspond to homologues of Cdc6, ORC4 and ORC3. PBANKA_110290 contained an AAA ATPase domain and a HMM profile analysis indicated that its closest homologue in human and yeast was Cdc6. Similarly, the closest homologue of PBANKA_134880 was ORC4 in human and yeast. Finally, homologues of PBANKA_051390 were only found in *Plasmodium* and *Babesia* but encoded a PF07034 PFAM domain corresponding to the N-terminus of ORC3. We then determined the relative abundance of MCM5-3xHA in chromatin-enriched NaCl fractions between non-activated and 15 second-activated gametocytes. Western blot analysis showed that 30% of MCM5-3xHA was present in the high salt fraction in non-activated gametocytes. This ratio almost doubled 15 s after activation but remained unchanged in the presence of 1294 (*Figure 2D*). Altogether, this suggests that loading of the MCM2-7/Cdt1 complex onto ORC1-5/Cdc6 complex happens around 15 s after activation of gametogenesis and requires active CDPK4.

## Myristoylation of CDPK4 is required to initiate DNA replication but not for axoneme activation

CDPK4 has previously been shown to be myristoylated in *P. falciparum* (*Wright et al., 2014*). We thus interrogated whether myristoylation could differentially regulate CDPK4 functions. To this aim, we generated a line in which the myristoylated glycine 2 was replaced by an alanine residue preventing myristoylation (*Figure 3—figure supplement 1* part 1). As for CDPK4-KO parasites, CDPK4$^{G2A}$-2xmyc microgametocytes did not exflagellate while a CDPK4-2xmyc control was not impaired (*Figure 3A*). Consistently, more than 85% of CDPK4$^{G2A}$-2xmyc and CDPK4-KO gametocytes remained haploid while 28% of CDPK4-2xmyc gametocytes became octoploid. However 65% showed spindle-like structures showing that CDPK4 myristoylation is important to initiate the first round of DNA replication but nor for mitotic spindle assembly (*Figure 3A*).

Western blots of CDPK4-2xmyc revealed a large band in non-activated gametocytes close to the expected size of 63.1 kDa and a second band migrating around 55 kDa (*Figure 3B*), a pattern that was conserved during gametogenesis (*Figure 3—figure supplement 1* part 2). In CDPK4$^{G2A}$-2xmyc extracts, the large isoform migrated slower. In some instances myristoylation has been shown to accelerate electrophoretic mobility (*Demetriadou et al., 2017*; *Zhao et al., 2011*) suggesting that the large isoform is myristoylated. Conversely, the 55 kDa band did not show a different mobility. Proteins can undergo additional post-translational covalent modifications with one or more palmitoyl groups after *N*-myristoylation. Using an acyl-RAC assay, we were however not able to detect significant evidence of CDPK4 palmitoylation (*Figure 3—figure supplement 1* part 3). To investigate how myristoylation could regulate CDPK4 function, we first determined its importance for protein localisation and protein-protein interactions. Immunofluorescence assays on CDPK4-2xmyc and CDPK4-3xHA, revealed that CDPK4 showed a broad cellular distribution in microgametocytes at 0 and 15 s after activation (*Figure 3—figure supplement 1* part 4). We were however not able to detect any obvious difference in CDPK4$^{G2A}$-2xmyc localisation at both time points. Similarly, immunoprecipitation of CDPK4-2xmyc and CDPK4$^{G2A}$-2xmyc did not reveal any significant changes in interacting partners under the tested conditions (*Figure 3—figure supplement 1* parts 5 and *Supplementary file 1*).

Myristoylation is also crucial to promote weak and reversible protein-membrane interactions. The CDPK4-2xmyc 55 kDa isoform was fully solubilised during hypotonic lysis while 90% of the CDPK4-2xmyc large isoform was solubilised during hypotonic lysis and fully released with a subsequent carbonate treatment (*Figure 3C*). Interestingly, the large isoform in CDPK4$^{G2A}$-2xmyc lysates was fully solubilised during hypotonic lysis. In parallel we also asked whether CDPK4 could be identified in a chromatin-enriched NaCl fraction. We found a minor population of myristoylated CDPK4-2xmyc in the NaCl fraction which was not observed for CDPK4$^{G2A}$-2xmyc. Altogether this highlights that a major population of CDPK4 is soluble whereas a minor population of the large CDPK4 isoform is associated with membranes and possibly with chromatin.

As the short CDPK4 isoform did not appear to be myristoylated, we hypothesised it may correspond to a shorter protein whose translation could be initiated at a second start codon. The only possible second start codon upstream the catalytic domain was methionine +57, which we replaced by a codon coding for an alanine residue (*Figure 3—figure supplement 1* part 1). The resulting CDPK4$^{M57A}$-2xmyc line only expressed the large isoform (*Figure 3D*) confirming that the translation of the 55 kDa isoform is initiated at a second start codon and cannot be myristoylated. Phenotyping of this line revealed that microgametocytes only expressing the large 63 kDa myristoylated isoform, progressed to the octoploid stage, assembled axonemes but showed a dramatic reduction in exflagellation compared to a control line (*Figure 3E*). Altogether this demonstrates that the large CDPK4 isoform is mainly required to initiate the first round of DNA replication while the short non-myristoylated isoform is essential to complete late gametogenesis (*Figure 3F*). Interestingly, either short or long CDPK4 isoforms seem to support mitotic spindle assembly.

## CDPK4 phosphorylates multiple proteins in gametocytes

To better understand how CDPK4 regulates these distinct biological processes we set out to identify its substrates using an analogue sensitive kinase (AS-kinase) engineered to contain a small gatekeeper residue (*Allen et al., 2007*). Bulky ATP analogues have been used to specifically label the targets of such modified kinases. The γ-phosphate of these artificial ATP-analogues is replaced with a

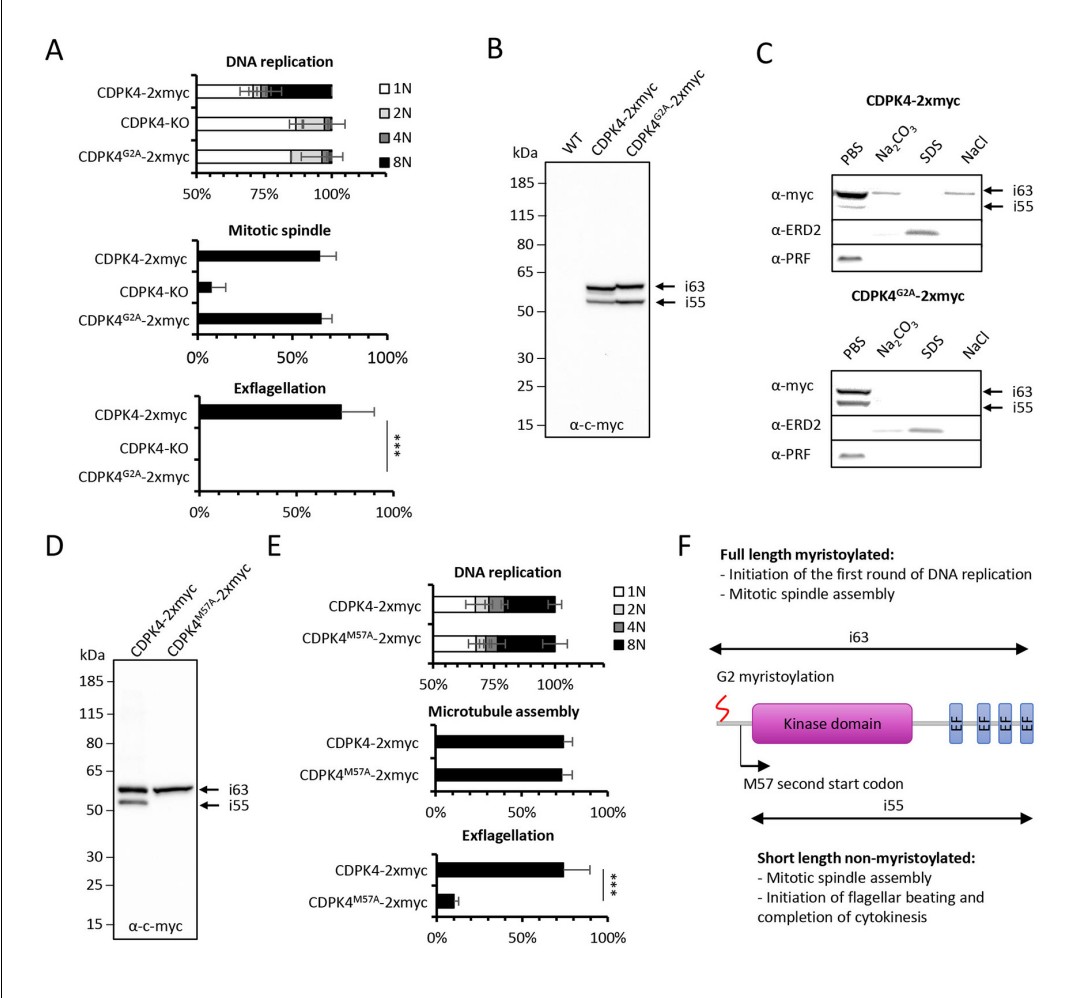

**Figure 3.** Myristoylation of CDPK4 is required to initiate the first round of DNA replication while non-myristoylated CDPK4 is important to complete gametogenesis. (A) Effect of *cdpk4* deletion or G2A substitution on microgametocyte DNA replication, mitotic spindle assembly, and exflagellation. G2A mutation mimics *cdpk4* deletion with ca. 70% of microgametocytes remaining at the 1N state and another 20% blocked at the diploid stage (n = 3). Consistently G2A substitution completely blocks exflagellation (n = 3) but does not prevent formation of mitotic spindles (n = 2). Error bars show standard deviations, *** Student's T-test, p≤0.001. (B) Western blot analysis showing expression of CDPK4-2xmyc and CDPK4$^{G2A}$-2xmyc proteins in complemented CDPK4-KO lines. Migration of the CDPK4$^{G2A}$-2xmyc isoform around 63 kDa (i63) is affected, suggesting it is myristoylated while the 55 kDa isoform (i55) migration is not affected suggesting it is not myristoylated. (C) CDPK4-2xmyc and CDPK4$^{G2A}$-2xmyc are both mainly detected in the PBS fraction. A minor population of CDPK4$^{G2A}$-2xmyc i63 is partly recovered in membrane-associated or NaCl fractions while the CDPK4$^{G2A}$-2xmyc i55 is fully solubilised in PBS. Antibodies against the ERD2 integral membrane protein and the soluble PRF protein were used as controls. (D) M57A substitution leads to expression of only i63, indicating that i55 originates from a second translation start at methionine 57. (E) A line expressing CDPK4$^{M57A}$-2xmyc shows a strong defect in exflagellation but is not affected in DNA replication and axoneme assembly indicating that the short non-myristoylable CDPK4 isoform is mainly required to control late gametogenesis but not to initiate the first round of DNA replication. Error bars show standard deviations, n = 3, *** Student's T-test, p≤0.001. (F) The two major CDPK4 isoforms and their identified roles during gametogenesis.

The following figure supplement is available for figure 3:

**Figure supplement 1.** CDPK4 myristoylation does not affect the kinase localisation and related protein-protein interactions.

thiophosphate and can then be used to purify the cognate substrates by chemical means. We first attempted to generate a line expressing a CDPK4$^{S147G}$-3xHA allele. As for CDPK4$^{S147M}$-3xHA, only a transient population having incorporated the S147G mutation and the 3xHA tag could be detected (data not shown). We were however able to clone a line in which endogenous *cdpk4* was replaced by a *cdpk4$^{S147G}$* allele (*Figure 4—figure supplement 1* part 1). This line showed a limited reduction

in exflagellation indicating that S147G substitution also imposes a slight fitness cost on CDPK4 function (*Figure 4—figure supplement 1* part 2).

To identify CDPK4 substrates, we compared incorporation of thiophosphate from *N6*-phenylethyl ATP-γS in lines expressing *cdpk4*$^{S147G}$ and *cdpk4*$^{S147M}$ alleles, respectively. Data processing with MaxQuant and Proteome Discoverer allowed to detect a total of 19 phosphorylated peptides in CDPK4$^{S147G}$ lysates only, ten of which were found to interact with CDPK4 (*Figure 4A* and *Supplementary file 2*). Among these proteins, we decided to name those without predicted function SOC, for substrate of CDPK4. Ten proteins were indeed conserved *Plasmodium* proteins of unknown function and one protein, SOC2, was annotated as cyclin-related protein 2 but it was suggested be unrelated to cyclins (*Roques et al., 2015*). Intriguingly, we also identified GAP40, a protein expressed in the merozoite stage (*Otto et al., 2014*) and important for the inner membrane complex biogenesis in *Toxoplasma gondii* (*Harding et al., 2016*).

Target identification with AS-kinases relies on cell lysates as a source of substrates and would identify potential substrates of CDPK4 implicated in regulation of both early and late

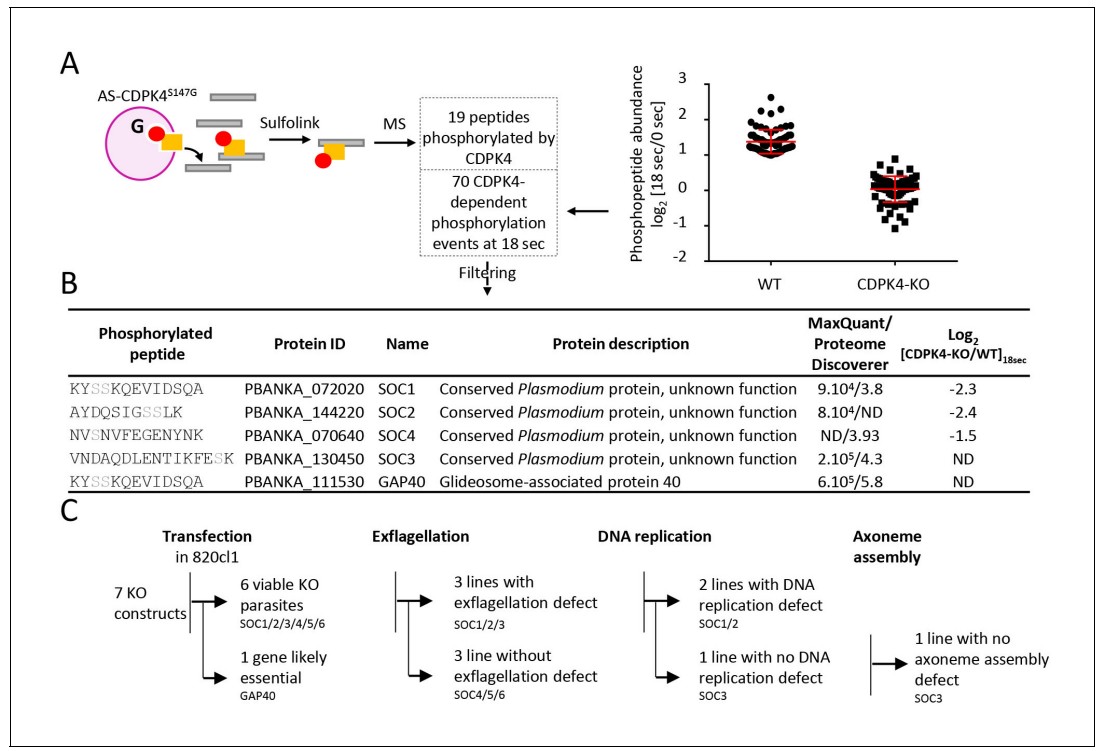

**Figure 4.** Identification of CDPK4-dependent phosphorylation during the first seconds of gametocyte activation. (**A**) Analogue sensitive-CDPK4$^{S147G}$ uses *N6*-phenylethyl ATP-γS (Red) to thiophosphorylate its substrates while the CDPK4$^{S147M}$ allele that cannot accommodate *N6*-phenylethyl ATP-γS is used as a control. Thiophospho-tryptic peptides are captured by SulfoLink Coupling Resin and submitted to LC-MS/MS analysis. Nineteen phosphopeptides were found to be thiophosphorylated by AS-CDPK4$^{S147G}$. Phosphoproteomic analysis revealed that 70 phosphopeptides on 61 proteins showed a 2-fold increase at 18 s post XA activation in the WT but not in the CDPK4-KO line. Data are from two biological replicates. (**B**) Among these latter, SOC1, SOC2 and SOC4 are phosphorylated by AS-CDPK4$^{S147G}$ and could represent early effectors of CDPK4. Conversely, GAP40 and SOC3 were not differentially phosphorylated in the CDPK4-KO line but were robustly identified as phosphorylated by AS-CDPK4$^{S147G}$ and could represent CDPK4 effector to control late gametogenesis. List of corresponding peptides and description of their proteins, MaxQuant intensity/ Proteome Discoverer Xcorr, and log$_2$[CDPK4-KO/WT]$_{18sec}$ phosphorylation ratio; sites assigned as phosphorylated by the two search engines are highlighted in grey. (**C**) GAP40 could not be deleted in asexual stages, suggesting an essential role in erythrocytic asexual multiplication. Knocking-out SOC4 or the SOC5 and SOC6 controls did not produce any defect in exflagellation nor DNA replication while SOC1, 2 and 3 KO lines showed a strong reduction in exflagellation. For SOC1-KO and SOC2-KO lines a defect in DNA replication was observed while SOC3-KO was not impaired in DNA replication and axoneme assembly.

The following figure supplement is available for figure 4:

**Figure supplement 1.** Identification and preliminary characterisation of CDPK4 substrates.

gametogenesis. To identify physiologically relevant substrates in early gametogenesis, we took advantage of the high synchronicity of early gametogenesis to profile the phosphoproteome in a CDPK4-KO line and its wild-type counterpart at 0 and 18 s after XA activation, when CDPK4 is required to initiate DNA replication and mitotic spindle assembly. In total we were able to identify 10,580 phosphopeptides mapping on 1,734 *P. berghei* proteins (*Supplementary file 3*). Among those, we identified 70 phosphopeptides mapping on 61 proteins that showed a 2-fold upregulation in the WT between 0 and 18 s but not in the CDPK4-KO line (*Figure 4A*). Among the 19 peptides phosphorylated by AS-CDPK4$^{S147G}$, those mapping on SOC1, SOC2 and SOC4 showed a 2-fold upregulation or more at 18 s in the WT but not in the CDPK4-KO line. This suggests that these three proteins are likely to represent CDPK4 physiological substrates to regulate early gametogenesis and were retained for further analysis (*Figure 4B*). Conversely, the remaining substrates identified with the AS-kinase could represent CDPK4 effectors controlling late gametogenesis. Among those, further analysis was focused on SOC3 and GAP40 proteins that were identified with the highest confidence based on both MaxQuant and Proteome Discoverer analyses (*Figure 4B*).

## The three roles for CDPK4 during gametogenesis are mediated by three distinct substrates

To investigate the roles of SOC1 to 4 and GAP40 during gametogenesis, we attempted to individually knock-out their encoding genes (*Figure 4B and C*, *Figure 4—figure supplement 1* parts 3 to 6). As controls, we included *soc5* and *soc6* (*Figure 4—figure supplement 1* parts 7 and 8) that were only detected by either MaxQuant or Proteome Discoverer but not differentially phosphorylated in the CDPK4-KO mutant. For *gap40,* no transgenic parasites could be obtained and this gene was not retained for further analysis. Viable KO parasites could be detected in mixed asexual stages for six genes and non-clonal populations were first assessed for microgametocyte DNA replication and exflagellation (*Figure 4—figure supplement 1* parts 9 and 10). No defects were observed in the SOC4-KO line as well as in the SOC5-KO and SOC6-KO control lines suggesting that these proteins do not represent crucial CDPK4 effectors in the regulation of male gametogenesis. Defects in DNA replication or exflagellation were observed for SOC1-KO, SOC2-KO, and SOC3-KO lines that were cloned for in depth characterisation.

SOC1-KO line showed a dramatic reduction in exflagellation and 50% of microgametocytes remained haploid (*Figure 5A*). As SOC1 phosphorylation requires CDPK4 activity within the first 20 s it is likely that SOC1 is a CDPK4 effector to control loading of the pre-replicative complex. Consistently co-immunoprecipitation of SOC1-3xHA in non-activated gametocytes identified MCM proteins and CDPK4 as abundant interactors (*Figure 5B*). As observed for CDPK4-3xHA and CDPK4-2xmyc, SOC1-3xHA did not show a significant enrichment in the nucleus but rather showed a diffuse cytoplasmic localisation (*Figure 5B*) raising the possibility it may also have other functions during male gametogenesis. Interestingly, 40% of SOC1-KO microgametocytes reached the octoploid state suggesting this protein is not essential to initiate the next two rounds of DNA replication. Although no domain could be identified in SOC1, this protein was found to be conserved in eukaryotes (*Figure 5—figure supplement 1* part 1). Eukaryotic homologues contained SAPS domains that have been found to be involved in $G_1$ to S-phase transitions by associating to SIT4 phosphatase (*Luke et al., 1996*). SOC1 is also expressed in asexual blood stages and its deletion was associated with a slight fitness cost in these stages (*Figure 5—figure supplement 1* part 2) suggesting its role is conserved in multiple stages of the *Plasmodium* life cycle.

The SOC2-KO line showed a dramatic reduction in exflagellation. Unlike SOC1-KO, proportions of haploid microgametocytes in 10 min activated microgametocytes were similar in the KO and its WT counterpart indicating that SOC2 is not required for initiation of DNA replication and DNA replication per se (*Figure 5C*). However, 47% of the parasites remained diploid and 20% and 16% reached the 4N and 8N levels, respectively. SOC2 could thus represent a CDPK4 effector required to complete mitosis in order to enter the subsequent round of replication. This prompted us to image mitotic spindles in the SOC2-KO line one minute after activation. As opposed to the 820cl1 parental line, we were rarely able to observe incorporation of α-tubulin into distinct mitotic spindles at 1 min post-activation, a phenotype similar to what was observed when control parasites were treated with 1294 prior to activation by XA (*Figure 5C*). At 10 min post-activation, SOC2-KO showed normal axonemes encircling the nucleus (*Figure 5C*). This 623 kDa protein appeared to be specific to *Plasmodium* (*Figure 5—figure supplement 1* part 1) and is predicted to encode six

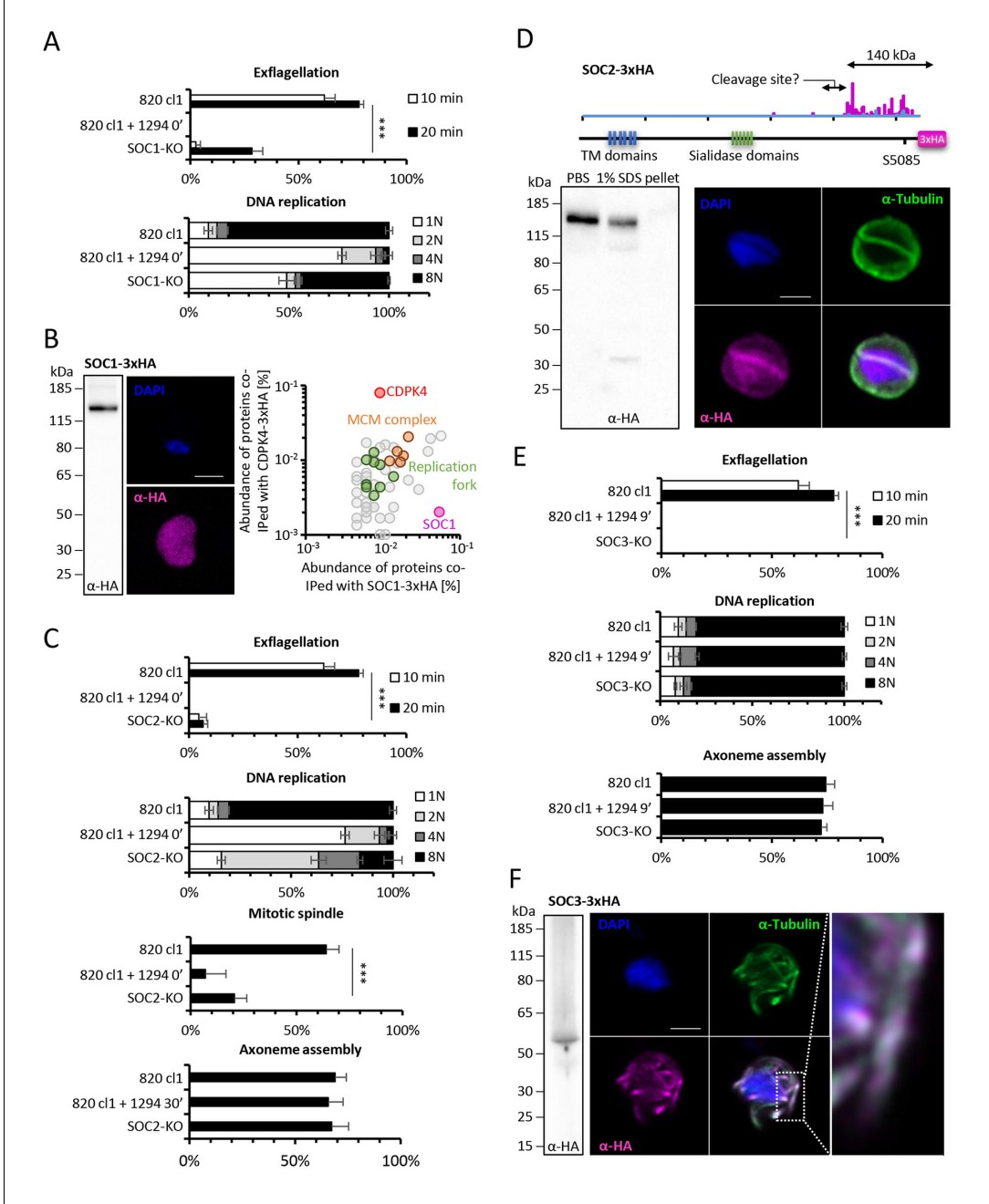

**Figure 5.** Functional characterisation of three identified substrates of CDPK4. (**A**) A SOC1-KO line shows a strong reduction in exflagellation (n = 3). SOC1 is required only for the 1N/2N transition as parasites reaching the 2N level were able to further progress to the octoploid state (n = 2). Error bars show standard deviations, *** Student's T-test, p≤0.001. (**B**) SOC1-3xHA shows a diffuse cytoplasmic distribution (scale bar = 2 μm) but interacts with CDPK4 and proteins of the MCM complex. (**C**) SOC2-KO line is strongly impaired for exflagellation (n = 3). SOC2 is not required for the 1N/2N transition but for each following transition indicating it is not involved in the initiation of DNA replication nor for DNA replication per se but most likely during the three successive endomitoses (n = 2). Consistently, a strong reduction in parasites exhibiting mitotic spindles at 1 min post-activation is observed in the SOC2-KO line (n = 2). However no defect in axoneme assembly could be detected (n = 2). Error bars show standard deviations, *** Student's T-test, p≤0.001. (**D**) SOC2 is predicted to be a 645 kDa polytopic protein. Immunoprecipitation of SOC2-3xHA recovers peptides covering the last 1707 a.a. only; corresponding unique spectral counts are indicated in pink. Similarly co-immunoprecipitation of CDPK4 only recovered peptides covering the last 1137 a.a. of SOC2; corresponding unique spectral counts are indicated in blue. The C-terminal fragment colocalises with mitotic spindle α-tubulin but not with peripheral α-tubulin. Scale bar = 2 μm. (**E**) SOC3 is not required for DNA replication (n = 2) and axoneme assembly (n = 2) but is essential for exflagellation (n = 3). Error bars show standard deviations, *** Student's T-test, p≤0.001. (**F**) SOC3 colocalises with axonemal α-tubulin in exflagellating gametes. Scale bar = 2 μm.

*Figure 5 continued on next page*

*Figure 5 continued*

The following figure supplement is available for figure 5:

**Figure supplement 1.** Characterisation of three identified substrates of CDPK4.

transmembrane domains and six sialidase penultimate C-terminal domains. C-terminal 3xHA tagging of SOC2 only allowed detection of a c.a. 140 kDa protein, 70% of which was extracted in PBS. Interestingly, this protein contained the peptide phosphorylated by CDPK4. Immunoprecipitation of SOC2-3xHA recovered peptides mapping on the last 1137 amino acids suggesting SOC2 is processed into at least two different isoforms. The 3xHA tagged C-terminal end of SOC2-3xHA was found to colocalise with mitotic spindles further suggesting a role in their assembly (*Figure 5D*). Similarly to SOC1, SOC2 is also expressed in asexual blood stages and its deletion was associated with a strong fitness cost in these stages (*Figure 5—figure supplement 1* part 2) suggesting a conserved role in mitotic spindle assembly during the atypical schizogony of *Plasmodium* parasites.

Finally, SOC3-KO clone did not exflagellate but defects neither in DNA replication nor in axoneme assembly could be detected (*Figure 5E*) suggesting it represents a crucial effector of CDPK4 to complete cytokinesis. Concordantly, SOC3-3xHA was found to partially colocalise with axonemal α-tubulin in exflagellating microgametes (*Figure 5F*). As opposed to SOC1 and 2, SOC3 is mainly expressed in the male gametocyte (*Otto et al., 2014*) and deletion of its coding gene was not associated with any growth defect of asexual blood stages (*Figure 5—figure supplement 1* part 2). Phylogenetic analysis indicated that SOC3 is restricted to *Plasmodium* species suggesting that the molecular function of SOC3 is highly specific to control the activation of male gametocyte axonemes.

## Discussion

Kinases represent important regulators of *Plasmodium* development and play crucial roles for the asexual proliferation in erythrocytes, in liver cells and during transmission stages (*Brochet et al., 2015*; *Solyakov et al., 2011*; *Tewari et al., 2010*). However little is known about how kinases regulate progression of the life cycle of malaria parasites. By taking advantage of the highly synchronised nature of *P. berghei* gametogenesis we were able to pinpoint when and how CDPK4 controls multiple cell cycle events during this biological process (*Figure 6*).

In response to ingestion by the mosquito, male gametocytes undergo three rounds of DNA replication alternating with three endomitotic divisions to generate a cell with an 8N genomic complement. In eukaryotes, developmentally programmed polyploidy involves the selective loss of mitotic cyclin-dependent kinases (CDKs) activity and bypassing many of the processes of mitosis (*Zielke et al., 2013*). In this study, we have found that CDPK4 represents a pivotal kinase to enter S phase in response to mosquito ingestion by promoting the recruitment of the MCM2-7/Cdt1 complex onto origins of replication. We identified SOC1 as an important effector of CDPK4 in the first twenty seconds of activation for the 1N/2N transition. Although this protein does not contain recognisable domains, conserved eukaryotic homologues of SOC1 contain a SAPS domain (Sit4-associated proteins). In yeast, SAPS proteins function in $G_1$ to promote timely DNA replication by regulating G1 cyclin transcription (*Luke et al., 1996*). Given the time frame of microgametogenesis it is possible that SOC1 regulates initiation of DNA replication through different mechanisms in *Plasmodium*.

Genome replications alternate with three endomitotic divisions. We found that SOC2, a second protein that is phosphorylated by CDPK4 within 20 s of activation, is not necessary to reach the 2N level but is important to progress to the 4N state, suggesting this early substrate is not involved in DNA replication but is required to complete the first mitosis. Accordingly, SOC2 localised at the mitotic spindles during mitosis and deletion of its coding gene was associated with marked defects in mitotic spindle formation. It is likely that SOC2 represents a *Plasmodium*-specific microtubule-associated protein required for chromosome segregation by controlling mitotic spindle assembly during the atypical cell cycle of *Plasmodium* parasites. Interestingly most SOC2-KO microgametocytes did not reach the 4N state suggesting a checkpoint might be operational to ensure that the second genome replication is initiated only after successful completion of the first mitosis. This is in

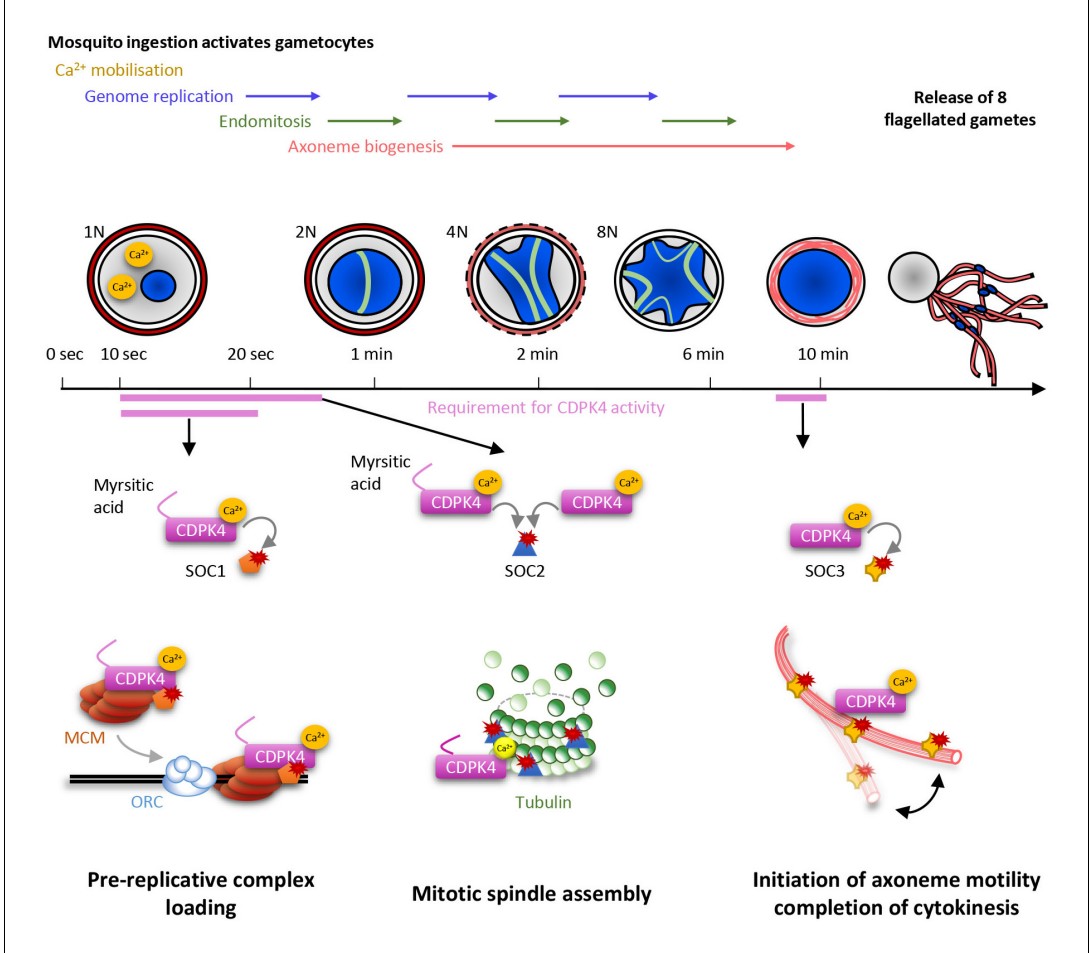

**Figure 6.** Model showing the roles of CDPK4 during *Plasmodium* male gametogenesis. Activation of male gametocytes by a drop in temperature and xanthurenic acid leads to mobilisation of intracellular calcium within a lag phase of ten seconds. Calcium activates a myristoylated isoform of CDPK4 which activity is required during the next ten seconds to facilitate the loading of the MCM2-7/Cdt1 complex onto ORC1-5/Cdc6 complex requiring SOC1. The subsequent assembly of mitotic spindles also requires (i) CDPK4 activity within the 30 fist second of gametogenesis and (ii) SOC2 which is a microtubule-associated protein phosphorylated by CDPK4 within the twenty fist seconds of gametogenesis. Between 30 s and 9 min, CDPK4 activity is neither required for the three successive rounds of genome replication and endomitosis nor for the assembly of the eight axonemes. Finally, activity of a non-myristoylable isoform of CDPK4 is required for the initiation of axoneme motility via the CDPK4 substrate SOC3.

agreement with recent findings showing that down-regulation of the cyclin-related kinase 4 leads to defects in both DNA replication and mitotic spindle formation during schizogony of *P. falciparum* (*Ganter et al., 2017*). Interestingly, we show here that regulation of DNA replication and mitotic spindle assembly are simultaneously controlled by a single kinase suggesting that, in *Plasmodium*, some S- and M-phase events may be simultaneously co-regulated. However, previous findings showed that small molecules affecting mitotic spindle stability did not impair DNA replication in asexual erythrocytic stages and in microgametocytes (*Billker et al., 2002*; *Sinou et al., 1998*) and it remains unclear how cell cycle transitions are regulated in *Plasmodium* parasites. On the other hand, we were able to observe fully formed axonemes in SOC2-KO gametocytes indicating that once initiated, axoneme assembly proceeds independently from some cell cycle checkpoints. Further studies will be required to investigate how S- and M-phase transitions are regulated during the atypical cell cycle of *Plasmodium* microgametes.

It is only during the final steps of gametogenesis that the eight genomes are incorporated into the eight exflagellating haploid gametes. This is achieved when the forming gamete swims out of the residual gametocyte body and drags a haploid genome that remains attached to a mitotic

spindle pole. Here, we have found that CDPK4 activity is also required to initiate axoneme motility and hence, to complete cytokinesis. We have linked this late role to an axoneme associated protein, SOC3. Its deletion did not impair genome replication nor axoneme assembly but blocked axoneme activation. Protein phosphorylation and dephosphorylation is known to be important to initiate axoneme motility but the cause-and-effect relationship between phosphorylation and motility remains poorly understood (*Wirschell et al., 2011*). From this study, we propose that SOC3 represents a *Plasmodium*-specific phosphoprotein required to initiate axoneme motility and consequently cytokinesis during microgametogenesis.

Deletion of *soc1* and *soc2* did not block the 1N/2N and 2N/4N transition as drastically as CDPK4 inhibition. This strongly suggests that other identified or non-identified CDPK4 substrates cooperate with SOC1 and SOC2 to load the pre-replicative complex and assemble mitotic spindles. In addition, multiple kinases and phosphatases have been shown to control male gametogenesis (*Guttery et al., 2012*; *Tewari et al., 2010*) and changes in phosphorylation levels between 0 and 18 s after activation highlighted a broad phospho-dependent response. It is thus likely that a plastic phosphorylation-dependent network underlies regulation of the initiation of DNA replication and mitotic spindle assembly and the exact molecular roles of SOC1 and SOC2 and of their phosphorylation remains to be elucidated. Conversely, deletion of *soc3* completely blocked exflagellation suggesting that SOC3 may be a focal component for the activation of axoneme motility.

During microgametogenesis, CDPK4 activity is essential only during two short time windows: within 10 to 30 s after activation and seconds prior to exflagellation. This highlights that CDPK4 is likely integrating multiple signals to be active at the right time and place. During the first window, the kinase is activated by a PKG-dependent $Ca^{2+}$ mobilisation that occurs ten seconds after XA stimulation (*Brochet et al., 2014*). Myristoylation of the large CDPK4 isoform is another signal required to initiate the first round of DNA replication but not to initiate axoneme motility. Interestingly, both large myristoylated and short non-myristoylated isoforms seemed to support mitotic spindle assembly. This indicates that myristoylation per se is not essential for CDPK4 activity *in vivo* in the absence of the 56 amino acids upstream of methionine +57. *In vitro*, full length non-myristoylated recombinant CDPK4 proved to be an active enzyme suggesting myristoylation is important for *in vivo* regulation (*Ojo et al., 2012*). We were however not able to provide evidence that CDPK4 myristoylation is required for specific cellular targeting nor for protein-protein interactions. Intriguingly, global profiling of protein myristoylation in human revealed that a large part of *N*-myristoyltransferase substrates are localised to the nucleus and mediate nuclear processes (*Thinon et al., 2014*). Collectively, this suggests that the additional 56 amino acids upstream the second start codon not only serves as an extra sequence to accommodate N-terminal myristoylation but may also represent a regulatory domain to control CDPK4 nuclear functions.

CDPK4 is regarded as a potential target for a transmission blocking strategy and proof of principle that chemical inhibition of CDPK4 by bumped kinases inhibitors such as 1294 can block *Plasmodium* transmission *in vivo* has been demonstrated (*Ojo et al., 2012*). It is likely that CDPK4 strict essentiality during malaria transmission is explained by its multiple functions identified in this study. Bumped kinase inhibitors such as 1294 appear to have a broad-spectrum anti-protozoal activity by targeting functional orthologues of CDPK4 (*Van Voorhis et al., 2017*) and this study provides a first insight into the molecular functions of this atypical $Ca^{2+}$-dependent protein kinase. Remarkably, CDPK4 has been previously reported to be important for sporozoite motility (*Govindasamy et al., 2016*) and oocyst formation (*Billker et al., 2004*), two processes unrelated to male gametogenesis. Similarly, in the related apicomplexan parasite *Toxoplasma gondii*, the functional orthologue of CDPK4 (TgCDPK1) is essential for host cell egress and invasion (*Lourido et al., 2010*). It thus likely that this kinase not only represents a pleiotropic regulator of the cell cycle during gametogenesis but also a broad spectrum regulator of multiple other biological processes in the Apicomplexa.

## Materials and methods

### Parasite maintenance and transfection

*P. berghei* strain ANKA (*Vincke et al., 1966*) derived clone 2.34 (*Billker et al., 2004*), clone 820cl1 (*Mair et al., 2010*) and derived transgenic lines (*Supplementary file 4*) were maintained in CD1 outbred mice. The parasitemia of infected animals was determined by methanol fixed and Giemsa

stained thin blood smears. Female mice were used for all experiments. CD1 outbred mice were obtained from Charles River laboratories. Mice were specific pathogen free and subjected to regular pathogen monitoring by sentinel screening. They were housed in individually ventilated cages furnished with a cardboard mouse house and Nestlet. Mice were maintained at 21 ± 2°C under a 12 hr light/dark cycle and given commercially prepared autoclaved dry rodent diet and water *ad libitum*. Mice were used for experimentation at 6–9 weeks of age. All animal experiments were conducted with the authorisation numbers (GE/82/15 and GE/41/17) according to the guidelines and regulations issued by the Swiss Federal Veterinary Office.

For gametocyte production, parasites were maintained in mice phenyl hydrazine-treated three days before infection. One day after infection, sulfadiazine (20 mg/L) was added in the drinking water to eliminate asexually replicating parasites. Microgametocyte exflagellation was quantified three or four days after infection of mice by adding 4 µl of blood from a superficial tail vein to 70 µl exflagellation medium (RPMI 1640 containing 25 mM HEPES, 4 mM sodium bicarbonate, 5% FCS, 100 µM xanthurenic acid, pH 7.4). To calculate the number of exflagellation centres per 100 microgametocytes, the percentage of RBCs infected with microgametocytes was assessed on Giemsa-stained smears. For gametocyte purification, parasites were harvested in suspended animation (SA - RPMI1640 medium containing 25 mM HEPES, 5% FCS, 4 mM sodium bicarbonate, pH 7.20) and separated from uninfected erythrocytes on a Histodenz cushion made up from 48% of a Histodenz stock (27.6% w/v Histodenz -Sigma- in 5.0 mM Tris-HCl [pH 7.20], 3.0 mM KCl, 0.3 mM EDTA) and 52% SA with a final pH of 7.2. Gametocytes were harvested from the interphase.

Schizonts for transfection were purified from overnight cultures on a Histodenz cushion made up from 55% of a Histodenz stock and 45% PBS. Purified parasites were harvested from the interphase and centrifuged at 500 g for 3 min, resuspended in 100 µL Amaxa Basic parasite Nucleofector solution (Lonza, Switzerland) and added to 10–20 µg of precipitated DNA resuspended in 10 µl of $H_2O$. Cells were electroporated using the U-0033 program of the Amaxa Nucleofector II. Transfected parasites were resupended in 200 µl of fresh red blood cells and injected intraperitoneally into mice. Selection with 0.07 mg/mL pyrimethamine (Sigma-Aldrich, Switzerland) in drinking water (pH ~4.5) was initiated from day one post infection. Negative selection of CDPK4$^{S147M}$ and CDPK4-3xHA parasites expressing yFCU was performed through the administration of 5 fluorocytosine (1 mg/mL, Sigma) via the drinking water. Each mutant was genotyped using three combinations of primers, specific for either the WT or modified locus on both sides of the targeted locus (experimental designs are shown in Supplemental Figures). For allelic replacements, sequences were confirmed by Sanger sequencing using indicated primers. WT DNA controls were included in each genotyping panel. Lines were cloned when indicated. CDPK4 mutants and derivatives and MCM-3xHA parasites were generated in the 2.34 background and all other transgenics were generated in the 820cl1 background.

## Generation of targeting constructs

3xHA tagging, knock-out, and allelic replacement constructs were generated using phage recombineering in *Escherichia coli* TSA bacterial strain with PlasmoGEM vectors (http://plasmogem.sanger.ac.uk/). The vector used to tentatively knock-out *gap40* was PbGEM-335107. For final targeting vectors not available in the PlasmoGEM repository, generation of knock-out and tagging constructs was performed using sequential recombineering and gateway steps as previously described (*Pfander et al., 2013*, *2011*). A list of oligonucleotides used in this study is available in *Supplementary file 5*. For each gene of interest (goi), the Zeocin-resistance/Phe-sensitivity cassette was introduced using oligonucleotides *goi* HA-F x *goi* HA-R and *goi* KO-F x *goi* KO-R for 3xHA tagging and KO targeting vectors, respectively. Substitutions of CDPK4$^{S147}$ gatekeeper residue were introduced with a two-step strategy using λ Red-ET recombineering as described in (*Brochet et al., 2014*). The first step involved the insertion by homologous recombination of a Zeocin-resistance/Phe-sensitivity cassette flanked by 5' and 3' sequences of the codon of interest, which is amplified using the *cdpk4*-del147F x *cdpk4*-del147R primer pair. Recombinant bacteria were then selected on Zeocin. The recombination event was confirmed by PCR and a second round of recombination replaced the Zeocin-resistance/Phe-sensitivity cassette with a PCR product containing the S147M or S147G substitution amplified using *cdpk4* S147M mut-F or *cdpk4* S147G mut-F with *cdpk4* S147 mutR primer pairs, respectively. Bacteria were selected on YEG-Cl kanamycin plates. Mutations were confirmed by

sequencing vectors isolated from colonies sensitive to Zeocin with primers *cdpk4*-seq1F to *cdpk4*-seq4F. The modified library inserts were then released from the plasmid backbone using NotI.

Constructs to generate CDPK4-2xmyc, CDPK4$^{G2A}$-2xmyc, and CDPK4$^{M57A}$-2xmyc parasites were derived from plasmid p150 (*Billker et al., 2004*). For CDPK4-2xmyc, CDPK4$^{G2A}$-2xmyc targeting constructs, *cdpk4* open reading frame was amplified from genomic DNA with primers *cdpk4* WT-F x *cdpk4* WT-R and *cdpk4* G2A-F x *cdpk4* G2A-R, respectively. For CDPK4$^{M57A}$-2xmyc targeting construct, *cdpk4* open reading frame was amplified from genomic DNA with primers *cdpk4* M57A-1F x *cdpk4* M57A-1R and M57A-2F x *cdpk4* M57A-2R. Fragments were assembled by Gibson cloning into p150 plasmid previously digested with NheI and ApaI. *cdpk4* coding sequence was confirmed using primers *cdpk4*-seq1F to *cdpk4*-seq4F and GW1. Resulting sequences did not show any undesired mutations and plasmids were linearized at a unique HpaI site 689 bp upstream of *cdpk4* for insertion into the 5′ flanking sequence of the genomic *cdpk4* locus by a single crossover. Presence of the gene in transgenic parasites was confirmed by sequencing using primer *cdpk4*-seq1F to *cdpk4*-seq4F and GW1.

## Immunofluorescence labelling

Gametocytes immunofluorescence assays were performed as previously described (*Volkmann et al., 2012*). For HA, c-myc and α-tubulin staining, purified cells were fixed with 4% paraformaldehyde and 0.05% glutaraldehyde in PBS for one hour, permeabilised with 0.1% Triton X-100/PBS for 10 min and blocked with 2% BSA/PBS for 2 hr. Primary antibodies were diluted in blocking solution (rat anti-HA clone 3F10, 1:1000; polyclonal rabbit anti-c-myc ref C3956, 1:5000; mouse anti-α-tubulin clone DM1A, 1:1000, all from Sigma-Aldrich). Anti-rat Alexa594, anti-mouse Alexa488, anti-rabbit Alexa 488, Anti-rabbit Alexa594 were used as secondary antibodies together with DAPI (all from Life technologies, Switzerland), all diluted 1:1000 in blocking solution. Confocal images were acquired with a LSM700 or a LSM800 scanning confocal microscope (Zeiss).

## FACS analysis of gametocyte DNA content

DNA content of microgametocytes was determined by FACS measurement of fluorescence intensity of cells stained with Vybrant dye cycle violet (life Technologies). Gametocytes were purified and resuspended in 100 µl of SA. Activation was induced by adding 100 µl of modified exflagellation medium (RPMI 1640 containing 25 mM HEPES, 4 mM sodium bicarbonate, 5% FCS, 200 µM xanthurenic acid, pH 7.8). To rapidly block gametogenesis, 800 µl of ice cold PBS was added and cells were stained for 30 min at 4°C with Vybrant dye cycle violet and analysed using a Beckman Coulter Gallios 4. Microgametocytes were selected on fluorescence by gating on GFP positive microgametocytes when the 820cl1 or its derivatives were used. In this case ploidy was expressed as a percentage of male gametocytes only. When 2.34 or its derivatives were analysed, gating was performed on both micro- and macrogametocytes and ploidy was expressed as a percentage of all gametocytes. Per sample, >50.000 cells were analysed.

## Fractionation and immunoprecipitation

For cell fractionation, purified gametocytes were washed and resuspended in PBS, PBS and 1% Triton X-100, PBS and 1M NaCl, or PBS and 0.1 M Na$_2$CO$_3$ [pH 11.5]. Cells were lysed by freezing and thawing followed by sonication on ice. Pellet and soluble fractions were separated by centrifugation for 1 hr at 14,000 rpm at 4°C. The solubility of ERD2 (MRA-1 from bei resources) and PRF from (*Plattner et al., 2008*) were also assessed in the different conditions as controls. Co-immunoprecipitation of CDPK4-3xHA, CDPK4-2xmyc, MCM5-3xHA, SOC1-3xHA protein complexes were performed with gametocytes fixed for 10 min with 1% formaldehyde, lysed in RIPA buffer and the supernatant was subjected to affinity purification with magnetics beads conjugated with anti-HA antibody (Roche) or anti-c-myc antibody (Sigma). A WT control was included in parallel and proteins for which we recovered peptides in the WT control were not retained for further analysis.

In CDPK4-3xHA interaction experiments at 0 and 15 s post-activation, 1 µl/mL of benzonase nuclease was added prior to immunoprecipitation. Magnetic beads were then resuspended in 50 µl of 1x NuPAGE LDS sample loading buffer containing 5 mM TCEP and incubated at 70°C for 10 min. Alkylation was carried out by addition of 2 mM iodoacetamide and incubation at room temperature in the dark for 30 min. Samples were separated by polyacrylamide electrophoresis, stained with

colloidal Coomassie and processed for mass spectrometry analysis as previously described (*Pardo et al., 2010*), with each gel lane excised into eight slices. Samples were re-dissolved in 0.5% FA before LC -MS/MS analysis. Samples were re-dissolved in 0.5% FA before LC -MS/MS analysis on LTQ Orbitrap Velos coupled with Ultimate 3000 RSLCnano system. Peptides were loaded and desalted on a peptide trap (Acclaim PepMap C18, 100 μm i.d. x 20 mm, 100 Å, 5 μm) at 10 μl/min and then separated on a nano-analytical column (Acclaim PepMap C18, 75 μm i.d. x 250 mm, 100 Å, 3 μm) at a linear gradient of 4–32% ACN/0.1% FA in 60 min (cycle time 95 min) at 300 nl/min. The HPLC, mass spectrometer and columns were all from ThermoFisher (United Kingdom). The Orbitrap mass spectrometer was operated in the standard 'top 15' data-dependant acquisition mode while the preview mode was disabled. The MS full scan was set at m/z 380–1600 with the resolution at 30,000 at m/z 400 and the lock mass at 445.1200 and AGC at $1 \times 10^6$ with a maximum injection time at 200 msec. The 10 most abundant multiply-charged precursor ions, with a minimal signal above 3000 counts, were dynamically selected for CID fragmentation (MS/MS) in the ion trap, which had an AGC set at 5000 with the maximum injection time at 100 msec. The dynamic exclusion duration time was set for 45 s with ±10 ppm exclusion mass width. Raw data was processed with Proteome Discoverer 1.4 and searched with Mascot (MatrixScience) against a combined mouse and *P. berghei* ANKA database (P.bergheiANKA.proteome, v2 May 2015) with the following parameters: trypsin/P as enzyme, maximum of 2 missed cleavages, 10 ppm parent ion mass tolerance, 0.5 Da fragment ion mass tolerance, and variable modifications of oxidized M, carbamidomethyl C, deamidated NQ, Gln to pyroGlu (N-terminal Q), N-terminal acetylation and N-terminal formylation. Database search results were refined using Mascot Percolator (significance threshold <0.05, FDR < 1%). High confidence peptides were apportioned to proteins using Mascot Protein Family summary. Protein identification required at least three high-confidence unique peptide (FDR < 1%).

For all other interaction experiments, magnetic beads were suspended in 100 μl of 6 M Urea in 50 mM ammonium bicarbonate (AB). To this solution, 2 μl of DTT (50 mM in LC-MS grade water) were added and the reduction was carried out at 37°C for 1 hr. Alkylation was performed by adding 2 μl of iodoacetamide (400 mM in distilled water) during 1 hr at room temperature in the dark. Urea concentration was lowered to 1M with 50 mM AB, and protein digestion was performed overnight at 37°C with 15 μL of trypsin Promega (0.2 μg/μl). After beads removal, the sample was desalted with a C18 microspin column (Harvard Apparatus), dried under speed-vacuum, and re-dissolved in $H_2O$/$CH_3CN$/FA 94.9/5/0.1 before LC-ESI-MS/MS analysis. LC-ESI-MS/MS was performed on a Q-Exactive Hybrid Quadrupole-Orbitrap Mass Spectrometer (Thermo Fisher Scientific, United Kingdom) equipped with an Easy nLC 1000 system. Peptides were trapped on an Acclaim pepmap100, C18, 3 μm, 75 μm x 20 mm nano trap-column and separated on a 75 μm x 500 mm, C18, 2 μm Easy-Spray column. The analytical separation was run for 90 min using a gradient of $H_2O$/FA 99.9%/0.1% (solvent A) and $CH_3CN$/FA 99.9%/0.1% (solvent B). The gradient was run as follows: 0–5 min 95% A and 5% B, then to 65% A and 35% B in 60 min, then to 10% A and 90% B in 10 min, and finally stay at 10% A and 90% B for 15 min. The entire run was at a flow rate of 250 nL/min. ESI was performed in positive mode. For MS survey scans, the resolution was set to 70000, the ion population was set to $3 \times 10^6$ with a maximum injection time of 100 ms and a scan range window from 400 to 2000 m/z. For MS2 data-dependent acquisition, up to fifteen precursor ions were selected for higher-energy collisional dissociation (HCD). The resolution was set to 17500, the ion population was set to $1 \times 10^5$ with a maximum injection time of 50 ms and an isolation width of 1.6 m/z units. The normalized collision energies were set to 27%. Peak lists were generated from raw data using the MS Convert conversion tool from ProteoWizard. The peaklist files were searched against the Plasmodium Berghei_ANKA database (*Plasmodium* Genomic Resource, release 28, 5076 entries) using Mascot search engine (Matrix Science, London, UK; version 2.5.1). Trypsin was selected as the enzyme, with one potential missed cleavage. Fragment ion mass tolerance was set to 0.020 Da and parent ion tolerance to 10.0 ppm. Variable amino acid modification was oxidized methionine and fixed amino acid modification was carbamidomethyl cysteine. The mascot search was validated using Scaffold 4.7.3 (Proteome Software). Protein identifications were accepted if they could be established at greater than 95.0% probability and contained at least three unique identified peptides.

## Acyl-resin-assisted capture of S-acylated proteins.

Purified CDPK4-3xHA gametocytes were lysed in 20 mM Hepes, 1 mM EDTA, 1.5% triton X-100 and protease inhibitor at pH 7.4. S-acylated proteins were purified by Acyl-Resin-Assisted Capture as

previously described (*Forrester et al., 2011*). Briefly, following cell lysis, free sulfhydryl groups are blocked with 1.5% methyl methanethiosulfonate, and the lysate is then treated with the nucleophile hydroxylamine to cleave thioester bonds, with the subsequent capture of proteins containing free cysteine thiolates on thiopropyl sepharose beads. Samples with and without hydroxylamine (-HA and +HA) were loaded on a SDS-PAGE gel for western blot analysis. α-HA antibodies are used to detect CDPK4-3xHA, α-PRF was used as a negative control. *Toxoplasma gondii* tachyzoites lysates together with a α-TgGAP45 (*Frénal et al., 2010*) were used as a positive control for S-acylated protein enrichment.

## Purification of thiophosphorylated peptides

Purified gametocytes were activated with exflagellation medium for 10 s and washed twice with ice-cold PBS. Cell were resuspended in 100 μl lysis buffer (20 mM HEPES pH7.5, 150 mM NaCl, 1% NP40, protease phosphatase inhibitors) and lysed on ice for 30 min. Lysates were then incubated at 37°C in 1.6 ml of kinase buffer (20 mM HEPES pH7.5, 10 mM MgCl$_2$, 500 μM N$^6$-PhEt-ATPγS, 3 mM GTP, 200 μM ATP, protease phosphatase inhibitors and 10 μM CaCl$_2$). After 30 min EDTA was added to a final concentration of 20 mM. Samples were then centrifuged at 20,000 g for 10 min at 4°C and the supernatant was kept on ice. The insoluble pellet was resuspended in 500 μl of 8 M urea, 100 mM TEAB, protease/phosphatase inhibitor and sonicated. The remaining lysate was centrifuged at 14,000 rpm for 5 min at 4°C.

The soluble and urea extracts were combined and 3.6 mg of total proteins were precipitated with chloroform/methanol to remove detergents. Pellets were solubilised and denatured in 8 M urea/100 mM TEAB/45 mM TCEP at 37°C for 15 min and further diluted to just under 4 M urea with 100 mM TEAB/9 mM TCEP. Proteins were digested with Lys-C (1:300, Wako) at 37°C for 2 hr and further diluted to 1 M urea with 100 mM TEAB/9 mM TCEP and digested with trypsin (1:30, Thermo Fisher) at 37°C for 14 hr. Peptides were desalted on C18 Sep-Pak cartridges (Waters) and dried in Speed-Vac, then resolubilised in 20 mM HEPES buffer pH 7.0/50% ACN/1 mM TCEP. The thiophosphopeptides capture used SulfoLink Coupling resin as described in (*Hertz et al., 2010*). Briefly, the SulfoLink Coupling resin was washed with 20 mM HEPES pH 7.0/50% ACN (binding buffer) then blocked with BSA in the dark for 10 min. The resin was incubated with peptides overnight at 21°C in the dark, then washed sequentially for ten minutes with 50% ACN, H$_2$O, 5M NaCl, 50% ACN, 5% FA, 10 mM DTT and H$_2$O. Samples were then treated with 1 mg/ml oxone (Sigma) for 2 min (twice) to elute the bond thiophosphopeptides which were converted to phosphopeptides in this step. The eluted peptides were desalted in SDB-XC tip. Phosphopeptides were enriched on TiO$_2$ tip (Thermo Fisher) and sequentially eluted with 1.5% NH$_3$ and 5% pyrrolidine (Sigma). Both eluates were treated with 10 mM TCEP for 10 or 15 min, respectively, and acidified before LC-MS/MS analysis on a LTQ Orbitrap Velos coupled with Ultimate 3000 RSLCnano system (Thermo Fisher) as for CDPK4-3xHA samples with the following differences: peptides were separated at a linear gradient for 95 min, the mass spectrometer was operated in the standard top 15 data dependent acquisition mode, and ions with a signal above 3000 counts were selected for CID fragmentation. Raw data was processed in Proteome Discoverer 1.4 with both SequestHT and Mascot search engines. The dynamic modifications set in Mascot were Acetyl (Protein N-term), Deamidated (NQ), Iodo(Y), Phospho (STY) and Oxidation (M) with fragment mass tolerance set at 0.8 Da, while in SequestHT the selected dynamic modifications were Deamidated (NQ), Iodo(Y), Phospho (STY) with fragment ions mass tolerance set at 0.5 Da. Precursor mass tolerance was set at 20ppm for both. The peptide list was filtered with Percolator where the q-value was set at 0.01, and the search results were merged. In the AS-CDPK4$^{S147G}$ samples, peptides with an XCorr and an IonScore of more than 3.8 or 35, respectively, were considered as positive. The raw data were also processed by MaxQuant (version 1.5.2.8) with most of the parameters settings at default value except the following parameters: trypsin with maximum two missed cleavages sites; Oxidation (M), Deamidation (NQ) and Phospho (STY) were set as variable modification while no fixed modifications were set on Carbamidomethyl (C). Cut off of 0.75 and 4.0 were retained for the localisation probability and score difference. In the AS-CDPK4$^{S147G}$ samples, peptides with an intensity >7.5×10$^4$ were considered as positive.

## Phosphoproteome profiling

Activated and no-activated purified parasites were snap frozen in liquid nitrogen. Cells were lysed in 4% SDS, 50 mM NaCl, 100 mM Tris buffer (pH 7.4), 5 mM EDTA, 40 mM TCEP and Halt phosphatase and proteinase inhibitor (2x, Thermo Fisher), and heated at 95°C for 10 min, then processed by ultrasonic probe for 20 s (1s on, 1 s off) at 40% power. Samples was centrifuged at 14,000 rpm for 30 min then supernatant was saved. 300 µg of total protein was taken and alkylated with 20 mM IAA at RT for 45 min and precipitated with MTBE (methyl tert-butyl ether). The pellet was digested with 4 µg trypsin in 100 mM TEAB at 37°C. After two hours of incubation, 4 µg of trypsin were added and samples were incubated for another five hours. 80 µg of the tryptic digest was taken and labelled with TMT10plex (Life Technologies) in 300 mM HEPES buffer (pH 8.0). Samples were pooled together and SpeedVac dried. The mixed peptides were fractionated on a 4.6 mm i.d. x 250 mm XBridge BEH C18 column (130 Å, 3.5 µm, Waters) at pH 10 over a linear gradient of 5–35% ACN/ 0.1% $NH_3$/60 min cycle time. Fractions were collected every 30 s in a 96 well plate and pooled into 12 fractions.

Enrichment for phosphopeptides were then performed by immobilised affinity chromatography (IMAC) chromatography with PHOS-Select Iron Affinity Gel (Sigma-Aldrich) and then $TiO_2$ tips (Thermo Fisher). All procedures followed s manufacturer's instruction with some modifications. 100 µl of 50% suspension of PHOS-Select Iron Affinity Gel was used for each fraction. The peptides were redissolved in 50% $CH_3CN$/250 mM acetic acid/0.1% TFA (trifluoroacetic acid) (binding solution) and then added to the prewashed beads. After binding at room temperature with end-to-end rotation for 30 min, the beads were washed three times with the binding solution and once with $H_2O$. Phosphopeptides were eluted twice with 100 µl of 1.5% NH3/25% ACN and then dried in SpeedVac. The flow through and the first wash of IMAC beads were collected and dried in SpeedVac, and then phosphopeptides were enriched with $TiO_2$ tips. Phosphopeptides were eluted from the tip with 1.5% $NH_4OH$ and then 5% pyrrolidine. Both eluates were pooled, acidified and desalted on Graphite Spin Columns (Thermo) as instructed by the manufacturer's protocol.

IMAC and $TiO_2$ enriched phosphopeptides were redissolved in 0.5% FA before LC-MS/MS analysis. Peptides were loaded on a trap column (Acclaim PepMap C18, 100 µm i.d. x 20 mm) then separated on a 75 µm i.d. x 500 mm column (Acclaim PepMap C18) with a linear gradient of 4–32% ACN/0.1% FA for 120 min and a total of 150 min per cycle. The Orbitrap Fusion was operated at a Top 15 method. The MS full scan was in Orbitrap (m/z 380–1500) with a lock mass at 445.120025, a resolution at 120,000 at m/z 200, and an AGC at $4 \times 10^5$ with a maximum injection time at 50 msec. The 15 most abundant multiply-charged precursor ions (2+ to 6+), with a minimal signal above 50,000 counts, were dynamically selected for HCD fragmentation (MS/MS) and detected in Orbitrap with a resolution at 50,000 at m/z 200, and the AGC at $1 \times 10^5$ with the maximum injection time at 105 msec. The isolation width was 1.2 Da in quadrupole, and the collision energy was set at 38%. The dynamic exclusion duration time was set for 60 s with ±10 ppm exclusion mass width.

Raw data were processed in Proteome Discoverer 2.1 with both SequestHT and Mascot search engines against a combined protein database of *Plasmodium berghei* (P.bergheiANKA.proteome, v2 May 2015) and mouse (UniprotKB). The dynamic modifications set in both Mascot were Acetyl (N-term), Deamidated (NQ), Phospho (STY) and Oxidation (M), while in SequestHT Camabidomethyl (C) was set as a fixed modification. Settings were: Precursor mass tolerance at 20 ppm, fragment at 0.5 Da, and TMTplex as fixed modification. The search results were merged and the peptide list was filtered with Percolator where the q-value was set at 0.01, and the phosphorylation sites were localised by *phosphoRS* implemented in the PD2.1 with site probability at 0.75 as cut-off. Both unique and razor peptides were used for protein quantification, and the reporter abundances were based on S/ N, then the abundances was normalised on Total Peptide Amount, and the scaled with On Channels Average (per file). The co-isolation threshold was set at 50% to reduce the isolation interference.

The mass spectrometry proteomics data have been deposited to the ProteomeXchange Consortium via the PRIDE partner repository with the dataset identifier PXD005884.

## Protein sequence analysis

PFAM domains were searched using SMART (*Letunic et al., 2015*) (http://smart.embl-heidelberg. de/) and protein sequence similarity searches were performed using HMMER (*Finn et al., 2011*) (http://hmmer.org/).

## Acknowledgements

This work was supported by the Swiss National Science Foundation (grant BSSGI0_155852 to MB). MB is an INSERM investigator. We are very grateful to Oliver Billker for his continuous support and for sharing the p150 plasmid. We would like to thank Wesley Van Voorhis and Kayode Ojo (University of Washington) for sharing compound 1294, the PlasmoGEM team (Wellcome Trust Sanger Institute) for providing the PlasmoGEM vectors and Andy Waters (University of Glasgow) for sharing the 820cl1 line. We finally thank the proteomics, bioimaging and flow cytometry core facilities (Faculty of Medicine, University of Geneva) for technical assistance.

## Additional information

### Funding

| Funder | Grant reference number | Author |
| --- | --- | --- |
| Schweizerischer Nationalfonds zur Förderung der Wissenschaftlichen Forschung | BSSGI0_155852 | Mathieu Brochet |

The funders had no role in study design, data collection and interpretation, or the decision to submit the work for publication.

### Author contributions

HF, NK, BB, EH, Formal analysis, Investigation; LY, Formal analysis, Investigation, Methodology; MP, Formal analysis, Investigation, Writing—review and editing; JC, Investigation, Methodology, Writing—review and editing; MB, Conceptualization, Supervision, Funding acquisition, Investigation, Writing—original draft, Writing—review and editing

### Author ORCIDs

Lu Yu, http://orcid.org/0000-0001-8378-9112
Mercedes Pardo, http://orcid.org/0000-0002-3477-9695
Mathieu Brochet, http://orcid.org/0000-0003-3911-5537

### Ethics

Animal experimentation: All animal experiments were conducted with the authorization Number (GE/82/15 and GE/41/17) according to the guidelines and regulations issued by the Swiss Federal Veterinary Office.

## Additional files

### Supplementary files

• Supplementary file 1. Number of unique spectral counts detected for CDPK4, MCM5 and SOC1 interactors.

• Supplementary file 2. Peptides thiophosphorylated by AS-CDPK4$^{S147G}$.

• Supplementary file 3. Phosphoproteome profiling of 2.34 and CDPK4-KO lines at 0 and 18 s post-activation.

• Supplementary file 4. Parasite lines generated in this study.

• Supplementary file 5. Oligonucleotides used in this study.

## Major datasets

The following dataset was generated:

| Author(s) | Year | Dataset title | Dataset URL | Database, license, and accessibility information |
|---|---|---|---|---|
| Evelyn Hillner, Lu Yu, Mercedes Pardo, Jyoti Choudhary, Mathieu Brochet | 2017 | CDPK4 is a pleiotropic regulator controlling the atypical Plasmodium cell cycle during mosquito transmission | https://www.ebi.ac.uk/pride/archive/projects/PXD005884 | Publicly available at the EBI PRIDE database (accession no. PXD005884) |

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
