## [Decision Letter]

Thank you for submitting your article "CDPK4 is a pleiotropic regulator of Plasmodium transmission with short-lived activity" for consideration by *eLife*. Your article has been reviewed by three peer reviewers, and the evaluation has been overseen by a Reviewing Editor and Jonathan Cooper as the Senior Editor. The following individuals involved in review of your submission have agreed to reveal their identity: Markus Meissner (Reviewer #1); Pietro Alano (Reviewer #3).

The reviewers have discussed the reviews with one another and the Reviewing Editor has drafted this decision to help you prepare a revised submission.

Summary:

This is a comprehensive, elegant and well-conducted reverse/chemical genetics study on the biochemical functions of CDPK4, a Plasmodium calcium-dependent kinase that was previously shown to be essential for male gametogenesis in both *P. falciparum* and the rodent malaria parasite *P. berghei*. The authors use a gatekeeper mutant enzyme that is resistant to the 1294 inhibitor, providing an exquisite control for the assignment of any phenotype to the kinase itself. Interactomics / biochemical data pertaining to potential substrates are nicely complemented with a phosphoproteomics analysis. Using these approaches, the authors identify three distinct, temporally defined steps, where CDPK4 activity is required during gametogenesis, define the spatiotemporal resolution of CDPK4 action and provide a detailed mechanistic characterisation of the CDPK4 interaction partners and their role during these processes.

This is a very thorough and well performed characterisation of CDPK4 functions during sexual development. All experiments are well performed and most conclusions correspond to the presented data. This manuscript is a major step towards dissecting the activities of the pleiotropic CDPK4 enzyme in *P. berghei* male gamete activation. It is experimental soundness and originality together with the significance of the obtained results expand its interest beyond the field of malaria parasites.

Essential revisions:

1) It is a very complex manuscript, which would benefit from rearranging some of the figures/data. For example, the manuscript proposes distinct roles of CDPK4 in controlling an early and a late phase of the increase in ploidy. The supporting data derive from experiments, described in different parts of the manuscript, i) testing effects of the CDPK4 inhibitors at t=0 and 20 seconds from gamete activation, ii) analysing phenotypically the SOC-2 mutant and iii) investigating the morphology of mitotic spindles and axonemes in SOC-2 gametes and in 1294 treated wild type gametes. It is suggested to merge these parts to more clearly and coherently link the relevant supporting data. Further, it would be good to provide an overview table of all the different mutants, their purpose and phenotype (otherwise one is forced to switch constantly between main figures and supplement).

2) In the analysis of the CDPK4 -myc and -HA interactomes 10 minutes post activation an additional role of CDPK4 in flagellar beating, beyond the one in replication initiation, is emphasized, based on the overrepresentation of cytoskeleton/microtubule associated proteins in this interactome. However, these were top ranking proteins also in the above 15 second interactome (Figure 1), a finding that was not emphasized in the description of that experiment. The second argument to support the role of CDPK4 in flagellar beating was the comparatively lower abundance of replication associated proteins in this interactome compared to the ones described above, which raises similar questions as above as the total number of CDPK4 peptides in these experiments is significantly lower than those used for the analysis shown in Figure 1.

3) The text related to Figure 1 and Figure 3 states that the data in these figures depict association of MCMs and CDPK4 with chromatin. Such a conclusion would require additional data, e.g. the use of accepted marker for the chromatin fractions, and/or CHIP/chromatin immunoprecipitations. The data allow to conclude that tagged MCMs/ CDPK4 are enriched in specific solubility fractions, but it seems to me that assign them to chromatin is a bit of an over-interpretation. The text around Figure 3 mentions only chromatin ("we found a minor population of myristoylated CDPK4-2xmyc in the chromatin fraction", while the Figure 3 legend, more accurately, says "a minor population of CDPK4^G2A^-2xmyc i63 is partly associated with membrane or chromatin fractions". The statement pertaining to chromatin association should be softened throughout, unless additional data are provided.

4) A general observation is that most of the key conclusions are based on abundance/relative abundance/enrichment of given gene products in the mass spectrometry analysis of the co-IP complexes. Mention of unique and razor peptides and of the normalization procedure are restricted to the fourth paragraph of the subsection “Phosphoproteome profiling” (Methodology) and [Supplementary-material SD1-data] does not indicate what numbers are listed (unique peptides?). A synthetic description of the rationale for protein quantification in this label-free proteomics approach would be expected also in the Results section.

5) The data to support the model that CDPK4 is needed to mediate MCM recruitment to bind the ORC/Cdt1 complex is based on the analysis of the three CDPK4 interactomes i) before gamete activation; ii) 15 seconds after activation with a functional CDPK4 and iii) with a 1294 inhibited CDPK4. How is the overrepresentation of some peptides in the CDPK4 sample at 15 seconds affected by the fact that the total number of peptides in this sample is higher (4000; with those for CDPK4 itself: 255) than that before activation (3879; CDPK4: 238) and in the 1294 inhibited CDKP4 (3415; CDPK4: 207)?

6)Subsection “CDPK4 activity is required for loading the MCM complex onto chromatin and the ORC”, first paragraph. "…three proteins correspond to Cdc6, ORC3 and ORC4; no ORC6 homologue was identified in the Plasmodium genome". The fact the three protein were not automatically annotated indicates that they do not conform to the relevant HMM profiles. Please provide the evidence (and the bioinformatics approaches used to generate it) that these three proteins are actually homologues of Cdc6, ORC3 and ORC4 (e.g. phylogenetic trees in Supplementary Information).

7) Subsection “CDPK4 activity is required for loading the MCM complex onto chromatin and the ORC”, first paragraph. "…suggest that CDP4 activity is required for loading MCM/Cdt onto ORC/Cdc6". The following sentence reads as an overstatement; the data show that CDPK4 can be co-IPed with Cdc6 and ORC proteins upon activation, but one cannot conclude from these data that "CDP4 activity is required for loading MCM/Cdt onto ORC/Cdc6". First, there are not data here that pertain to activity; second, an equally valid hypothesis is that CDPK4 is recruited after ORC/Cdc6 have been loaded onto chromatin. Both points are addressed in the following paragraph, which ends with a very similar statement. Consider deleting the first one ("This suggests that CDPK4 activity is required for the loading of the MCM2-7/Cdt1 complex onto ORC1-5/Cdc6 complex within 15 seconds following gametocyte activation by XA".

8) The authors generate functional and non-functional versions of CDPK4 and nicely demonstrate the role of CDPK4 during different steps using a gatekeeper mutant and convincingly demonstrate that CDPK4 acts during a ~15 second window. However, they then use the HA-tagged version to identify specific interaction partners, though this version appears to be not functional during this time. This needs to be discussed accordingly. Do the authors obtain similar results using the functional myc-tagged version?

9) The authors used AS-kinase to identify substrates of CDPK4. Are these substrates also interaction partners identified in the Co-IPs? Could the author discuss this in more detail?

10) Figure 2. The imaging (in general) needs to be substantially improved. Please indicate what exactly we are looking at.

11) Figure 2. As the authors acknowledge themselves, the identified proteins are not necessarily indicating direct interactions with CDPK4, especially since later (Figure 5). The primary targets were identified using AS-kinase. As such this panel is rather distracting from the core message of the study.

12) In the analysis of CDPK4 myristoylation, the mobility shift of the higher band of the CDPK4 G2A mutant in the Western Blot (Figure 3) does not seem sufficient to support the major conclusion that CDPK4 in *P. berghei* is myristoylated as in *P. falciparum*. In the first paragraph of the subsection “Myristoylation of CDPK4 is required for its nuclear functions but not to complete gametogenesis”, discussing these results, is very confusing. "In CDPK4^G2A^-2xmyc extracts, the large isoform migrated slower, as expected in the absence of myristoylation". Does it imply that myristoylation is known to accelerate electrophoretic mobility of proteins? If so, a reference would be useful. If not, the paragraph should be re-worded and the point made clearer.

13) The authors use a G2A mutant, which cannot be myristoylated. For the localisation analysis (Figure 3—figure supplement 1) they then use cross-reactive antibodies that appear to stain the surface. This will not lead to any conclusive results and therefore this analysis needs to be repeated with non-cross-reactive antibodies (or alternatively a different tagged version of CDPK4). Is CDPK4 also palmitoylated? On the same point, in the absence of any obvious mislocalisation of the CDPK4 G2A mutant protein in the microgametocyte compared to the wild type, putatively myristoylate enzyme, it could be interesting to observe if changes of localization occur (and differ between wild type and the G2A mutant) upon gamete activation.

14) Figure 4: The use of the AS-kinase is very elegant and one of the strength of this paper. The authors identify 19 peptides. Are any of the proteins in their interactome analysis also found in this approach.

15) Figure 5. Please provide a quantification of the imaging data.

---

## [Author Response]

Essential revisions:

1) It is a very complex manuscript, which would benefit from rearranging some of the figures/data. For example, the manuscript proposes distinct roles of CDPK4 in controlling an early and a late phase of the increase in ploidy. The supporting data derive from experiments, described in different parts of the manuscript, i) testing effects of the CDPK4 inhibitors at t=0 and 20 seconds from gamete activation, ii) analysing phenotypically the SOC-2 mutant and iii) investigating the morphology of mitotic spindles and axonemes in SOC-2 gametes and in 1294 treated wild type gametes. It is suggested to merge these parts to more clearly and coherently link the relevant supporting data. Further, it would be good to provide an overview table of all the different mutants, their purpose and phenotype (otherwise one is forced to switch constantly between main figures and supplement).

The ‘result’ flow in the current manuscript reflected the experimental progression of our analysis. In the light of the reviewers’ comments we agree this might be confusing and make it difficult to link the three phenotypes of CDPK4 inhibition. We have now grouped the description of the three identified roles of CDPK4 in Figure 1 and subsequently describe the interactomes in Figure 2. We have modified the text accordingly. As suggested by the reviewers we have added a supplemental table listing all the different mutants, their purpose and main phenotype.

2) In the analysis of the CDPK4 -myc and -HA interactomes 10 minutes post activation an additional role of CDPK4 in flagellar beating, beyond the one in replication initiation, is emphasized, based on the overrepresentation of cytoskeleton/microtubule associated proteins in this interactome. However, these were top ranking proteins also in the above 15 second interactome (Figure 1), a finding that was not emphasized in the description of that experiment. The second argument to support the role of CDPK4 in flagellar beating was the comparatively lower abundance of replication associated proteins in this interactome compared to the ones described above, which raises similar questions as above as the total number of CDPK4 peptides in these experiments is significantly lower than those used for the analysis shown in Figure 1.

We agree with the reviewers that the role of CDPK4 in axoneme activation cannot be ascertained solely on the abundance of cytoskeletal proteins in CDPK4 immunoprecipitates, and this was not our intention. The microtubule cytoskeleton is indeed enriched in most immunoprecipitates from gametocytes and may represent non-relevant or even non-specific interactions. To avoid any confusion we have removed the panel D of Figure 2. The number of total peptides that were recovered in these experiments were indeed lower as peptides were directly digested on the beads and not after gel separation. To avoid such bias, we only use relative protein abundance in our quantification (please see point 5 for more details).

3) The text related to Figure 1 and Figure 3 states that the data in these figures depict association of MCMs and CDPK4 with chromatin. Such a conclusion would require additional data, e.g. the use of accepted marker for the chromatin fractions, and/or CHIP/chromatin immunoprecipitations. The data allow to conclude that tagged MCMs/ CDPK4 are enriched in specific solubility fractions, but it seems to me that assign them to chromatin is a bit of an over-interpretation. The text around Figure 3 mentions only chromatin ("we found a minor population of myristoylated CDPK4-2xmyc in the chromatin fraction", while the Figure 3 legend, more accurately, says "a minor population of CDPK4^G2A^-2xmyc i63 is partly associated with membrane or chromatin fractions". The statement pertaining to chromatin association should be softened throughout, unless additional data are provided.

We agree with the reviewers that high salt fraction could not be used as a synonym for chromatin fraction we have now rephrased the text accordingly and toned down our conclusions.

4) A general observation is that most of the key conclusions are based on abundance/relative abundance/enrichment of given gene products in the mass spectrometry analysis of the co-IP complexes. Mention of unique and razor peptides and of the normalization procedure are restricted to the fourth paragraph of the subsection “Phosphoproteome profiling” (Methodology) and [Supplementary-material SD1-data] does not indicate what numbers are listed (unique peptides?). A synthetic description of the rationale for protein quantification in this label-free proteomics approach would be expected also in the Results section.

The enrichment of molecular components identifies which GO terms are over-represented in a particular gene set which we now mention in the text. The relative abundance of each protein was calculated by dividing the number of spectral counts for each protein by the number of total spectral counts in the respective immunoprecipitate. We now indicate this in the relevant figure legends and have included the description of the numbers listed in [Supplementary-material SD1-data].

5) The data to support the model that CDPK4 is needed to mediate MCM recruitment to bind the ORC/Cdt1 complex is based on the analysis of the three CDPK4 interactomes i) before gamete activation; ii) 15 seconds after activation with a functional CDPK4 and iii) with a 1294 inhibited CDPK4. How is the overrepresentation of some peptides in the CDPK4 sample at 15 seconds affected by the fact that the total number of peptides in this sample is higher (4000; with those for CDPK4 itself: 255) than that before activation (3879; CDPK4: 238) and in the 1294 inhibited CDKP4 (3415; CDPK4: 207)?

We are comparing the relative abundance of each protein across the different conditions. We calculate the relative abundance of each protein by dividing the number of spectral counts for each protein by the number of total spectral counts in the respective immunoprecipitate.

Due to technical reasons, the number of spectral count is indeed variable across the different conditions. However, for the vast majority of identified proteins, the relative abundance is similar between the three conditions as shown in Figure 2—figure supplement 1 with correlation coefficients of ca 0.95 between each condition. For example, CDPK4 represents 6.3, 6.1, and 6.0% of total spectral counts, at 0 sec, 15 sec + XA and 15 sec + XA + 1294, respectively. Similarly, the relative abundance of the MCM complex is similar across the three conditions representing 5.7, 6.0 and 5.4% at 0 sec, 15 sec + XA, and 15 sec + XA + 1294, respectively. However, the ORC/Cdc6 complex is relatively more abundant in activated gametocytes with 3.1% at 15 sec + XA compared with 1.0 and 1.6% at 0 sec, and 15 sec + XA + 1294, respectively.

6)Subsection “CDPK4 activity is required for loading the MCM complex onto chromatin and the ORC”, first paragraph. "…three proteins correspond to Cdc6, ORC3 and ORC4; no ORC6 homologue was identified in the Plasmodium genome". The fact the three protein were not automatically annotated indicates that they do not conform to the relevant HMM profiles. Please provide the evidence (and the bioinformatics approaches used to generate it) that these three proteins are actually homologues of Cdc6, ORC3 and ORC4 (e.g. phylogenetic trees in Supplementary Information).

The analysis sustaining our conclusion was indeed absent in the text. We now explain in more detail in the main text and in the Methods that HMM profiles and/or domain analysis strongly suggest that these three proteins likely correspond to Cdc6, ORC3 and ORC4. However, this would have to be experimentally confirmed. This also holds true for all the other components of the pre-replicative complex.

7) Subsection “CDPK4 activity is required for loading the MCM complex onto chromatin and the ORC”, first paragraph. "…suggest that CDP4 activity is required for loading MCM/Cdt onto ORC/Cdc6". The following sentence reads as an overstatement; the data show that CDPK4 can be co-IPed with Cdc6 and ORC proteins upon activation, but one cannot conclude from these data that "CDP4 activity is required for loading MCM/Cdt onto ORC/Cdc6". First, there are not data here that pertain to activity; second, an equally valid hypothesis is that CDPK4 is recruited after ORC/Cdc6 have been loaded onto chromatin. Both points are addressed in the following paragraph, which ends with a very similar statement. Consider deleting the first one ("This suggests that CDPK4 activity is required for the loading of the MCM2-7/Cdt1 complex onto ORC1-5/Cdc6 complex within 15 seconds following gametocyte activation by XA".

We have deleted the first occurrence and we agree that it is highly likely that the ORC complex is already bound to DNA in non-activated gametocytes. We tried to C-terminally tag ORC2 with a 3xHA epitope to test this hypothesis but such a modification prevented exflagellation (data not shown). However inhibition of CDPK4 activity by 1294 prevents the stronger interaction between CDPK4/MCM2-7/Cdt1 with ORC1-5/Cdc6 and the enrichment of MCM5 in the NaCl fraction pertaining to activity. We do agree that CDPK4 may have additional functions independent of its kinase activity and may, for example, also serve as a chaperone for the pre-replicative complex.

8) The authors generate functional and non-functional versions of CDPK4 and nicely demonstrate the role of CDPK4 during different steps using a gatekeeper mutant and convincingly demonstrate that CDPK4 acts during a ~15 second window. However, they then use the HA-tagged version to identify specific interaction partners, though this version appears to be not functional during this time. This needs to be discussed accordingly. Do the authors obtain similar results using the functional myc-tagged version?

We did not find evidence that the CDPK4-3xHA is not functional. We observed delayed exflagellation for this line. This suggests that although the enzyme activity may be affected it does not dramatically affect DNA replication. A stronger interaction at 15 sec compared with 0 sec was also obtained using a CDPK4-2xmyc line which does not show delayed exflagellation. We have now included this data in the text, [Supplementary-material SD1-data], and Figure 2—figure supplement 1.

9) The authors used AS-kinase to identify substrates of CDPK4. Are these substrates also interaction partners identified in the Co-IPs? Could the author discuss this in more detail?

Please see answer to point 14.

10) Figure 2. The imaging (in general) needs to be substantially improved. Please indicate what exactly we are looking at.

We have improved the imaging and now refer to the sub cellular structures of interest in the figure legend.

11) Figure 2. As the authors acknowledge themselves, the identified proteins are not necessarily indicating direct interactions with CDPK4, especially since later (Figure 5). The primary targets were identified using AS-kinase. As such this panel is rather distracting from the core message of the study.

We agree with the reviewers this panel could lead to over interpretation of the results. We have consequently removed panel D of Figure 2.

12) In the analysis of CDPK4 myristoylation, the mobility shift of the higher band of the CDPK4 G2A mutant in the Western Blot (Figure 3) does not seem sufficient to support the major conclusion that CDPK4 in P. berghei is myristoylated as in P. falciparum. In the first paragraph of the subsection “Myristoylation of CDPK4 is required for its nuclear functions but not to complete gametogenesis”, discussing these results, is very confusing. "In CDPK4^G2A^-2xmyc extracts, the large isoform migrated slower, as expected in the absence of myristoylation". Does it imply that myristoylation is known to accelerate electrophoretic mobility of proteins? If so, a reference would be useful. If not, the paragraph should be re-worded and the point made clearer.

In some cases it has been shown that myristoylation accelerates the electrophoretic mobility of a protein, see PMID 21841012, 28137759 for example. We have now included these references in the text. It will unfortunately be very difficult to investigate incorporation of radioactively labelled myristate in this model as *P. berghei* gametocytes are produced in vivo, and down-regulation of NMT with a tetracycline-inducible system strongly affected parasite replication and consequently gametocytogenesis (PMID 23245327).

13) The authors use a G2A mutant, which cannot be myristoylated. For the localisation analysis (Figure 3—figure supplement 1) they then use cross-reactive antibodies that appear to stain the surface. This will not lead to any conclusive results and therefore this analysis needs to be repeated with non-cross-reactive antibodies (or alternatively a different tagged version of CDPK4). Is CDPK4 also palmitoylated? On the same point, in the absence of any obvious mislocalisation of the CDPK4 G2A mutant protein in the microgametocyte compared to the wild type, putatively myristoylate enzyme, it could be interesting to observe if changes of localization occur (and differ between wild type and the G2A mutant) upon gamete activation.

We did not find evidence for palmitoylation of CDPK4 and have now added this observation to the manuscript. We have re-analysed the localisation of CDPK4-3xHA, CDPK4-2xmyc, and CDPK4^G2A^-2xmyc in microgametocytes at 0 and 15 sec. We have used a more diluted commercial polyclonal antibody (C3956 from Σ) that only led to a very faint background in WT microgametocytes. However, we have not been able to identify obvious differences between the different lines and the two time points. We have added this analysis in Figure 3—figure supplement 1.

14) Figure 4: The use of the AS-kinase is very elegant and one of the strength of this paper. The authors identify 19 peptides. Are any of the proteins in their interactome analysis also found in this approach.

Some of these substrates are also interacting partners of CDPK4 but were not the most abundant interactors. We have now included this in the Results section and in [Supplementary-material SD2-data].

15) Figure 5. Please provide a quantification of the imaging data.

We do thank the reviewers for highlighting this point. As a role in mitotic spindle assembly was only found late in our project we did not quantify this aspect. We have now quantified the percentage of cell showing mitotic spindle structures at 1 min for WT, WT + 1294, and SOC2-KO lines. We also took this opportunity to check the formation of mitotic spindles in CDPK4^G2A^-2xmyc at 1 min post-activation. As the CDPK4^M57A^-2xmyc mutant which only encodes for the myristoylable isoform was able to go through the first mitosis we expected the CDPK4^G2A^-2xmyc line to be defective for mitotic spindle assembly. However, the analysis revealed that CDPK4^G2A^-2xmyc mutant did not show any defect for this process indicating that both short and long isoforms could support mitotic spindle assembly. We have now included this finding in the Results and Discussion sections.